# Application of tritium in precipitation and baseflow in Japan: A case study of groundwater transit times and storage in Hokkaido watersheds

Maksym A. Gusyev[1], Uwe Morgenstern[2], Michael K. Stewart[2,3], Yusuke Yamazaki[1], Kazuhisa Kashiwaya[4], Terumasa Nishihara[4], Daisuke Kuribayashi[1], Hisaya Sawano[1] and Yoichi Iwami[1]

[1]International Centre for Water Hazard and Risk Management (ICHARM), Public Works Research Institute (PWRI), Tsukuba, Ibaraki 305-8516, Japan
[2]GNS Science, Avalon, Lower Hutt, 5011, New Zealand
[3]Aquifer Dynamics & GNS Science, PO Box 30368, Lower Hutt, New Zealand
[4]Civil Engineering Research Institute for Cold Region (CERI), PWRI, Sapporo, Japan

*Correspondence to*: M.A. Gusyev (gusyev55@pwri.go.jp; maksymgusyev@gmail.com)

**Abstract.** In this study we demonstrate the application of tritium in precipitation and baseflow to estimate groundwater transit times and storage volumes in Hokkaido, Japan. To establish the long-term history of tritium concentration in Japanese precipitation, we used tritium data from the global network of isotopes in precipitation and from local studies in Japan. The record developed for Tokyo area precipitation was scaled for Hokkaido using tritium values for precipitation based on wine grown at Hokkaido. Then tritium concentrations measured with high accuracy in river water from Hokkaido, Japan, were compared to this scaled precipitation record and used to estimate groundwater mean transit times (MTTs). Sixteen river water samples in Hokkaido were collected in June, July and October 2014 at twelve locations with altitudes between 22 and 831 m above sea level and catchment areas between 45 and 377 km$^2$. Measured tritium concentrations ranged between 4.07 ($\pm$0.07) TU and 5.29 ($\pm$0.09) TU in June, 5.09 ($\pm$0.09) TU in July, and between 3.75 ($\pm$0.07) TU and 4.82 ($\pm$0.07) TU in October. We utilized TracerLPM (Jurgens et al., 2012) for MTT estimation and introduced a Visual Basic module to automatically simulate tritium concentrations and relative errors for selected ranges of MTTs, exponential-piston ratios, and scaling factors of tritium input. Using the exponential(70%)-piston flow(30%) model (E70%PM), we simulated unique MTTs for seven river samples collected in six Hokkaido headwater catchments because their low tritium concentrations are not ambiguous anymore. These river catchments are clustered in similar hydrogeological settings of Quaternary lava as well as Tertiary propylite formations near Sapporo city. However nine river samples from six other catchments produced up to three possible MTT values with E70%PM due to the interference by the tritium from the atmospheric hydrogen bomb testing 5-6 decades ago. For these catchments we show that tritium in Japanese groundwater will reach natural levels in a decade, when one tritium measurement will be sufficient to estimate a unique MTT. Using a series of tritium measurements over the next few years with 3 year intervals will enable us to estimate the correct MTT without ambiguity in this period. These unique MTTs will allow estimation of groundwater storage volumes for water resources management during droughts and

improvement of numerical model simulations. For example, the groundwater storage ranges between 0.013 and 5.07 km$^3$ with saturated water thickness from 0.2 and 25 m. In summary, we emphasise three important points from our findings: 1) one tritium measurement is already sufficient to estimate MTT for some Japanese catchments, 2) the hydrogeological settings control the tritium transit times of subsurface groundwater storage during baseflow, and 3) in future one tritium
measurement will be sufficient to estimate MTT in most Japanese watersheds.

## 1 Introduction

Improved understanding of groundwater dynamics is needed to answer practical questions of water quality and quantity for groundwater discharges such as wells and streams. Knowing groundwater travel times allows us to pin-point possible sources of groundwater pollution from agricultural activities, while estimates of groundwater volumes in the subsurface are
needed for sustainable management of water resources in many countries (Granneman et al., 2000; McMahon et al., 2010; Gusyev et al., 2012; Stewart et al., 2012; Morgenstern et al., 2015). In Japan, there is a need for a robust and quick approach to quantify the subsurface groundwater volume as an important component of the water cycle due to the recently enacted "Water Cycle Basic Law" on March 2014 (Tanaka, 2014). In addition, subsurface groundwater storage plays an important role in contributing to flood river flows and providing much of the river water during droughts. However, the complex
groundwater dynamics are often difficult to characterize on a river basin scale due to the absence of subsurface information. Therefore, a common practice is to utilize numerical models with simplified representations of the complex groundwater dynamics for rainfall-runoff simulation in river catchments. For example, a distributed hydrologic model BTOP, which has been applied globally (Magome et al., 2015) and in many river basins for detailed flood and drought hazard quantification (Gusyev et al., 2015, 2016; Navarathinam et al., 2015; Nawai et al., 2015), is used to simulate groundwater flow components
using the exponential mixing model (EMM) with mean travel distance of groundwater flow (Takeuchi et al., 2008). At the river basin scale, a simple and yet robust tracer such as tritium ($^3$H) is needed to characterize the groundwater bodies as they are drained by surface water features.

Tritium has been instrumental in providing information on hydrologic systems and surface-groundwater interactions in river waters in the Southern Hemisphere, but tritium tracer studies are still scarce in the Northern Hemisphere rivers (Michel
2004; Michel et al., 2015; Harms et al., 2016). In the Southern Hemisphere, tritium measurements in river water have been commonly used to understand groundwater dynamics by determining groundwater transit time distributions and by constraining groundwater flow and transport models (Stewart et al., 2007; Gusyev et al., 2013, 2014; Morgenstern et al., 2010, 2015; Cartwright and Morgenstern, 2015; Duvert et al., 2016). Tritium is a part of the water molecule and migrates through the water cycle while being inactive except for radioactive decay. The half-life of 12.32 yrs allows us to quantify
water lag time in the subsurface of up to 200 yrs even with the natural levels of tritium concentrations in precipitation, but requires the most sensitive equipment to detect the small concentrations of tritium currently found in river water in the Southern Hemisphere (Morgenstern and Taylor, 2009). Low concentrations of tritium in precipitation are derived from

cosmogenic generation in the upper atmosphere and high tritium concentrations have been contributed by anthropogenic point-source pollutions such as atmospheric nuclear weapons testing, nuclear fuel reprocessing, nuclear power plant accidents, and industrial applications (Akata et al., 2011; Matsumoto et al., 2013; Tadros et al., 2014). The high bomb tritium contribution compared to the very low tritium in current precipitation is expected to cause a long-lasting ambiguity in the groundwater reservoirs for the Northern Hemisphere (Stewart et al., 2012), especially for Japan with its low tritium concentrations due to mainly low-tritium maritime precipitation but large contributions of bomb tritium from atmospheric nuclear weapons testing. This ambiguity can be resolved with time series sampling, especially for water younger than 20 yrs due to the still-remaining steep gradient in the tritium output. Both the past record of tritium concentration in precipitation and tritium measurements in river water are required for application of tritium for understanding river-aquifer dynamics in river basins of Japan and other countries.

In this study, we explore the use of tritium to characterise groundwater dynamics for the specific tritium conditions of Japan, with large contributions of bomb-tritium from the continent, but low natural tritium concentrations from low-tritium maritime precipitation. Twelve headwater catchments in Hokkaido were selected to test the methodology of subsurface volume characterization from estimated groundwater transit times (Małoszewski and Zuber, 1982) using high-precision tritium analyses in Hokkaido river water. Firstly, we examine tritium data from the global network of isotopes in precipitation (GNIP) to determine the continuous times-series of tritium in precipitation for the Tokyo area in Japan. This times-series is scaled for Hokkaido, Japan, using inferred information about local tritium in young infiltrating water. Secondly, river water samples from Hokkaido were collected in the headwater catchments during surveys in June, July and October 2014 and analysed at the GNS Science low-level tritium laboratory in New Zealand. Then, the estimated Hokkaido tritium record was utilized with the river water measurements to determine groundwater transit times using the convolution integral with the exponential-piston flow model (EPM). Finally, the mean transit times (MTTs) are utilized with river baseflow discharge to estimate the subsurface storage volumes of the selected catchments. In addition, we discuss the suitability for tritium dating of headwater catchments in Hokkaido and other Japanese river basins in the past, at present, and in the future from these MTTs, and suggest the requirements for future tritium monitoring.

**2 Approach**

Subsurface water volumes are estimated by multiplying baseflow river discharges by groundwater transit times simulated using the convolution integral with tritium. The approach is demonstrated in a schematic diagram of a river catchment that drains subsurface groundwater storage by a river network (Fig. 1a). Precipitation with known tritium concentrations infiltrates into the subsurface and recharges the subsurface reservoir that is drained by the stream network. River water samples have mixtures of quick runoff and groundwater flow with different travel times and hence tritium concentrations. Baseflow dominates river discharge during dry periods when a river water sample represents only a mixture of groundwater with different tritium concentrations. Using the convolution integral, we can estimate groundwater transit times by inputting

the long-term record of tritium in precipitation and comparing the output with the tritium in the river water at baseflow. The time-dependent tritium concentration $C_{out}(t)$ [TU] at the time of sampling $t$, is defined at the groundwater discharge point such as a river, stream or spring by the convolution integral (Małoszewski and Zuber, 1982):

$$C_{out}(t) = \int_{-\infty}^{t} C_{in}(\tau) \; e^{-\lambda(t-\tau)} g(t-\tau) d\tau \qquad (1)$$

where $C_{in}(t)$ [TU] is the input tritium concentration, $\lambda$ [yr$^{-1}$] is the tritium decay term of 0.056262 [yr-1], and g(t) [-] is the weighting (or system response) function that is a simplified representation of the complex groundwater pathways (see the cross-sectional diagram of the unconfined aquifer in Fig. 1b and the partially confined aquifer in Fig. 1c, Małoszewski and Zuber, 1982). The unconfined aquifer (Fig. 1b) is recharged over the entire length of the aquifer and is described by the exponential mixing model (EMM), which has only one fitting parameter (mean transit time MTT). In the partially confined

aquifer, the confined portion that does not receive recharge is represented by the piston flow model while the unconfined part of the aquifer is described by the EMM resulting in the exponential-piston flow model (EPM), see Fig. 1c. The system response function of the EPM is defined by Małoszewski and Zuber (1982):

$$g(t-\tau) = \frac{1}{Tf} \, exp(-\frac{(t-\tau)}{Tf} + \frac{1}{f} - 1) \qquad \text{for } t \geq T(1-f) \quad (2a)$$
$$g(t-\tau) = 0 \qquad \text{for } t < T(1-f) \quad (2b)$$

where T [yrs] is the mean transit time (MTT) of groundwater, and $f$ [-] is the ratio of the volume of the exponential component to the total volume of the aquifer that is equal to 1 for the EMM and is close to 0 for nearly piston flow. The convolution integral was evaluated using TracerLPM (Jurgens et al., 2012) that uses the EPM ratio, which is defined as n=1/f -1. The mobile groundwater volume of the subsurface reservoir, $V(t)$ [m$^3$], at time of sampling t at baseflow (Małoszewski and Zuber, 1982) is

$$V(t) = Q_b(t) * T = A(t)\bar{h}(t) \qquad (3)$$

where $Q_b(t)$ [m$^3$ yr$^{-1}$] is the baseflow river discharge, $A$[m$^2$] is the area of subsurface groundwater storage, $\bar{h}(t)$[m] is the average saturated groundwater thickness, which can be found as volume, $V(t)$, divided by the subsurface storage area, $A$. The baseflow discharge can be estimated using a baseflow separation method such as one introduced by Stewart (2015):

$$Q_b(t) = Q_b(t-1) + k + f_c * (Q(t) - Q(t-1)) \qquad \text{for Q(t) > Q}_b\text{(t-1) + k} \quad (4a)$$

$$Q_b(t) = Q(t) \qquad (4b)$$

where $Q_b$(t-1)[m$^3$ s$^{-1}$] is the baseflow at time t-1, $f_c$[-] is the constant fraction of the increase or decrease of the river discharge during an event, k[m$^3$ s$^{-1}$ hr$^{-1}$] is the slope of dividing line, and Q(t)[m$^3$ s$^{-1}$] and Q(t-1)[m$^3$ s$^{-1}$] are river discharges at time t and t-1, respectively. Using this estimated groundwater volume as initial condition we can estimate changes of the subsurface groundwater storage including low as well as high recharge conditions:

$$[V(t+1) - V(t)]/ \Delta t = R(t) - Q_b(t) \qquad (5)$$

where V(t+1)[m$^3$] and V(t)[m$^3$] is the groundwater volume of subsurface storage at time t+1 and t, $\Delta t$[s$^{-1}$] is the time interval and R[m$^3$ s$^{-1}$] is the groundwater recharge. From Eq. (5), the groundwater storage is depleted by river network drainage when R < Q$_b$ during periods of little or no groundwater recharge and is replenished during periods when R > Q$_b$.

# 3 Study area of the Hokkaido Island

## 3.1 Climatic conditions

The twelve headwater catchments investigated in this study are located in the western and central parts of Hokkaido Island (Fig. 2). Hokkaido Island is one of four main Japanese islands with most of its population centered in Sapporo city and is surrounded by the Sea of Japan on the west, the Sea of Okhotsk on the north, and the Pacific Ocean on the east (Fig. 2). It has the cool temperate climate of the Koeppen climate classification due to its location between the northern limit of the temperate climate and the southern limit of the cool temperate climate (JMA, 2016). Hokkaido weather patterns vary across the island with 30-year annual average precipitation in the following cities: 1.04 m in Muroran (Southwest), 1.11 m in Sapporo (West), 1.13 m in Rumoi (Northwest), 0.79 m in Abashiri (Northeast), 1.04 m in Kushiro (Southeast), and 1.04 m in Asahikawa, see Fig. 2 (JMA, 2016). The summer climate of Hokkaido Island is dictated by cold polar and warm northern Pacific air masses and does not have the distinct rainy season typical of other locations in Japan. For example, Sapporo city with 30-year monthly average precipitation of 0.05 m in June, 0.08 m in July and 0.12 m in August has drier summers than Tokyo with 30-year monthly average precipitation of 0.17 m in June, 0.15 m in July, and 0.17 m in August (JMA, 2016). In the summer season, the western climate zone of Hokkaido Island has fair weather for most of the period with daily mean temperature ranging from 15°C to 20°C. August is the hottest month of the year and the daily maximum temperature can reach 30°C at some inland places in the upper Ishikari River (JMA, 2016).The weather becomes unsteady and changeable from September due to the influence of typhoons and fronts, which makes September the wettest month of the year with 30-year monthly average precipitation of 0.14 mm. Air temperature decreases gradually towards the winter season while snowfall may occur in late September in the mountains of upper Ishikari River basin (Fig. 2). From late November on, the daily average temperature stays mostly below 0°C until the end of March. Cold air masses flow eastward in winter bringing freezing temperatures with heavy snowfalls to the central mountain ranges facing the Sea of Japan and clear skies to areas fronting the Pacific side. For Sapporo city, the 30-year monthly average temperature is -0.9 $^o$C in December, -3.6 $^o$C in January and -3.1 $^o$C in February resulting in the duration of continuous snow cover from late November to early April. February is the coldest month of the year with daily minimum temperature reaching -20°C in some inland places of the Ishikari River basin. On the Pacific side, the daily average temperature is slightly higher resulting in shorter duration of continuous snow cover from December to early March in Muroran and Kushiro cities. In Sapporo city, the winter 30-year monthly average precipitation of about 0.1 m in December- February accumulates as snow on the ground resulting in maximum snow depth of 0.05 m in December, 0.08 m in January, and 0.1 m in February. A large volume of snowfall results in thick snow cover on the ground staying throughout much of the winter season and preventing freezing of the soil (Iwata et al., 2010). This implies that water with tritium may infiltrate into the subsurface as groundwater recharge. From March, the 30-year daily average temperature sometimes goes up above 0°C in the plain area, and instead of snow, rain starts to fall, initiating the snowmelt process, which usually ends in early April in the low elevation areas and between May and June in the mountainous areas.

## 3.2 Topography

Out of twelve selected catchments, eleven are situated in the headwaters of the Ishikawa River basin, which is the third largest Japanese river basin with a drainage area of 14330 km$^2$, and one is situated in the Rumoi River basin with area of 270 km$^2$ (Fig. 2). The mean annual discharge of the Ishikari River is about 500 m$^3$ s$^{-1}$ at Sapporo city located on the western side of the Ishikari plain, which is the largest lowland plain of Hokkaido Island (Fig. 2), The topography of the Ishikari River basin varies from the Ishikari plain at the seashore of the Okhotsk Sea to 2290 meters above sea level (masl) at Mt. Asahi, which is located at the center and is the highest point of Hokkaido Island (Ikeda et al., 1998). The ridge of Mts. Ishikari and Mikuni extends to the northeast-southwest direction and is a surface water divide between the Ishikari, Tokachi and Tokoro Rivers flowing to the Japan Sea, Pacific Ocean, and Okhotsk Sea respectively (Hasegawa et al., 2011). The selected headwater catchments are surrounded by forested area within catchments of existing dams except Tougeshita (Table 1).

The eleven investigated catchments in the Ishikari River basin with areas between 45 and 377 km$^2$ are located at altitudes between 80 and 850 masl and have stream drainage densities from 10 to 16 km/km$^2$ and mean slopes between 0.19 and 0.31 (Table 1). Six of the investigated catchments share catchment boundaries and three are located in close proximity to each other (Fig. 2). For example, Otaruani (#1) and Takinosawa (#2) are neighbouring catchments and are located upstream of Jozankei Dam on the south side of the ridge that recharges the alluvial aquifer of Sapporo city. The Izariirisawa (#3) and Honryujyoryu (#4) are neighbouring catchments located upstream of Houheikyo Dam, and Kouryu (#5) and Hakusen (#6) neighbouring catchments located upstream of Kanayana Dam, situated on the western and eastearn side of the same surface water divide, respectively. The Rubeshinai (#9) and Ishikaridaira (#10) stations in the central part of Hokkaido Island are situated on two tributaries that drain headwaters of the Ishikari River to the downstream Taisetsu Dam Lake (Fig. 2). The Piukenai (#11) catchment is located upstream of Chubetsu Dam and its tributary drains the eastern side of Mt. Asahi. Tougeshita station is located at the lowest altitude of 22 masl in the Rumoi River basin; its catchment has the smallest slope of 0.16 and maximum elevation of 712 masl (Fig. 2). The outlets of these selected headwater catchments except Tougeshita are located upstream of existing dams and have operational Ministry Land Infrastructure Transport and Tourism (MLIT) river gauging stations (Table 1). These river gauges report historical and real-time hourly river water levels as well as inferred historical river discharges for some years (WIS, 2016). WIS (2016) provides historical and real-time precipitation and some precipitation gauges also report snow depth on the terrain surface. WIS (2016) also provides hourly precipitation, reservoir storage, and discharge at dam offices and estimated reservoir inflows.

## 3.3 Surface and subsurface geology

The geology of Hokkaido is divided into the eastern region (the Northeast Japan Arc), western region (the Kuril Arc), and central region in the arc-arc collision (Hasegawa et al., 2011). It has three distinct active Quaternary volcanic fields (Fig. 3, AIST, 2012): the south-west area of the Oshima belt, central area of the Hidaka belt (Taisetsu-Tokachi-Shikaribetu) and eastern area of the Nemuro belt (Akan-Shirekoto) (Hasegawa et al., 2011). In the south-west area, the irregular arrangement

of plains, mountains and volcanoes such as Shikotsu and Toya with large calderas and pyroclastic plateaus is different from the central and eastern regions (Hasegawa et al., 2011). The Ishikari plain is located to the west of the Yubari Mountains and is situated on top of a deep alluvial fan, which occurs in the low lying areas (AIST, 2012). The alluvial aquifer of the Ishikari plain has groundwater flow oriented towards the sea with recharge from the surrounding elevated low permeability

formations that are situated south of Sapporo city (Dim et al., 2002; Sakata and Ikeda, 2013). Following the arc-arc collision region, Hidaka, Yubari and Teshio mountain ranges cross Hokkaido Island from south to north (Hasegawa et al., 2011). The Teshio Mountains consist of Cretaceous-Tertiary folded formations while the Yubari Mountains have Jurrassic-Cretacious formations (Sorachi Group consisting of greenstone with several inclusions of basaltic pyroclastic lava, hyaloclastite and diabase, chert, micrite limestone, and sandstone with felsic tuff) and serpentinite in and around the main ridge (Hasegawa et

al., 2011). For the central volcanic field, Sounkyou and Taisetsu volcanoes are located in the Taisetsu mountain range, which is comprised of over 20 mountains including Mt. Asahi (Hasegawa et al., 2011). The Taisetsu volcano is located a few kilometres north-east of Mt. Asahi and has produced Plinian pumice-fall and pyroclastic flow deposits with a large eruption about 35 ky ago resulting in the Ohachidaira caldera.

The surface and subsurface geology of the twelve selected catchments is obtained from 1:50K geological maps of the

Hokkaido area as shown in Fig. 3a-f (AIST, 2012) and summarized in Table 1. Six catchments are located in the eastern geologic region and share similar geologic features of Tertiary propylite and Quaternary lava formations, see Fig. 3a and 3b (AIST, 2012). In Fig. 3a, the geology of the Otarunai (#1) and Takinosawa (#2) catchments is dominated by Tertiary propylite of Zenibako Group overlaid by andesite lavas (Fig. 3a) and is similar to the Hakusen (#6) catchment with propylite of Izarigawa Group that is overlaid by Quaternary lavas and Tertiary sandstones (Fig. 3b). The Quaternary volcanic lavas

with augite hypersthene andesite are dominant for the Izariirisawa (#3), Honryujyoryu (#4), and Kouryu (#5) catchments including propylite, quartz and shale for Kouryu (Fig. 3b). In Fig. 3c, the Okukatsura (#7) catchment is located in the Cretaceous geologic area, which is quite different from the other catchments, and includes sandy siltstone, siltstone and sandstone. The Ikutora (#8) catchment is described as rhyolitic welded tuff overlain by Quaternary volcanic lavas and underlain by metamorphic and igneous rocks (Fig. 3d). In Fig. 3e, a variety of geologic material is demonstrated in the

Taisetsu mountain range with dominant Quaternary lavas in three selected catchments including slate and sandstone of the Hidaka Group for Rubeshinai (#9), Pre-Tertiary slate for Ishikaridaira (#10) and the Sounkyou welded tuff for Piukenai (#11). The Tougeshita (#12) river catchment in the Rumoi River basin is dominated by Neogene mudstone, mudstone with interbedded sandstone, and Quaternary alluvial deposits near river channels (Fig. 3f). Exploratory bores drilled prior to construction of Chubetsu, Jyozankei and Kanayama dams showed aquifer materials ranging from highly permeable shallow

alluvial sand and gravel materials near river channels to low permeability underlying formations. The observed water levels demonstrated groundwater heads below the terrain surface in these bores.

### 3.3 Sampling at selected catchments and historical tritium

We selected one location for tritium sampling in June, July and October 2014 in each of 12 Hokkaido headwater catchments (Fig. 2 and 3). Sampling locations were visited in the June survey during the dry period and diurnal fluctuation of river water levels was observed due to snow melt. Water samples were collected only at 6 stations where river water levels and discharges were below mean annual flows provided in Table 1. 10 river water samples were collected excluding Okukatsura (#7) and Tougeshita (#12) in July 2014, but only one river sample was analysed due to a large rainstorm event started during the sampling trip. A sample of the rain was also collected at the Kogen hot spring situated at about 1200 masl (Fig. 2). In October, river water samples were collected by local dam officers at the 9 locations during cross-section measurements of river profiles when river water levels and flows were below normal. We obtained water level data in all stations except Kouryu (#5) and Hakusen (#6), which were washed away during an October flood, and estimated the river discharges as demonstrated for five stations in Fig. 4. For Izariirisawa and Honryujyoryu stations, we investigated the Hoheikyo dam inflow, which was about 5 $m^3 s^{-1}$ on October 23$^{rd}$ and was similar to the 6 $m^3 s^{-1}$ inflow of Hoheikyo dam, which is located in the neighbouring river catchment (Fig. 2). The Ikutora station had an erroneous record in February-March as indicated by an arrow in Fig. 4e. For five stations, we conducted baseflow separation using Eq. (4) with optimum values of $f_c$ and $k$ to estimate baseflow during sampling (Fig. 4): 5.64 $m^3 s^{-1}$ in June (#1a) and 3.66 $m^3 s^{-1}$ in October (#1b) at Otarunai, 0.48 $m^3 s^{-1}$ at Okukatsura (#7), 10.9 $m^3 s^{-1}$ in June (#8a) and 9.47 $m^3 s^{-1}$ in October (#8b) at Ikutora, 1.32 $m^3 s^{-1}$ in June (#9a) and 0.53 $m^3 s^{-1}$ in October (#9b) at Rubeshinai, and 0.27 $m^3 s^{-1}$ at Tougeshita (#12) (Fig. 4). Samples collected during below normal river discharges were analysed for tritium, deuterium (D), and oxygen-18 ($^{18}O$) by the tritium laboratory in New Zealand (Morgenstern and Taylor, 2009). Water chemistry of collected samples was analyzed at the laboratory of Forest Hydrology and Erosion Control Engineering, Graduate School of Agriculture and Life Sciences, University of Tokyo, Japan, including silica (Si) with the molybdenum yellow method. In a follow-up sampling trip, we collected one river water sample near Otarunai station during winter baseflow conditions on February 24$^{th}$, 2016. Accumulated snow layers of up to 3 m in the winter makes the access to rivers for sampling difficult in Hokkaido headwater catchments.

The long-term tritium record of Tokyo precipitation was constructed using the tritium records of the International Atomic Energy Agency (IAEA) stations and Japanese stations such as the National Institute of Radiological Sciences (NIRS) and Japan Chemical Analysis Center (JCAC). The GNIP Tokyo station, which was originally located at 36$^{o}$N, has a monthly tritium record of 18 yrs with samples being measured by the University of California from 1961 to 1963 and by the IAEA Vienna Laboratory from 1964 to 1979 (IAEA/WMO, 2014). The JCAC, located in Sanno near Tokyo, has recorded monthly tritium values in precipitation at the GNIP station in Japan from April 2007 to the present. In addition, tritium concentrations in precipitation were inferred from wine measurements in Kofu between 1952 and 1963 (Takahashi et al., 1969) and in Hokkaido from 1970 to 1994 (Ikeda et al., 1998) and used to estimate pre- and post-bomb period tritium in Japanese precipitation, respectively.

## 4 Results and Discussion

### 4.1 Tritium time-series in precipitation and recharge

Figure 5 shows the reference tritium input curve of Japanese precipitation between 1951 and 2015 developed for the Tokyo area and the scaled reference curve developed for Hokkaido recharge. The inferred annual tritium concentrations derived from Kofu and Hokkaido wine are indicated by triangles and circles, respectively. The Tokyo GNIP station values show a sharp decline from 1000 TU to 100 TU between 1964 and 1968, a small increase from 1969 to 1973 and a steady decline up to the end of the record in 1979. A similar increasing pattern between 1969 and 1973 is observed in the annual tritium data inferred from Hokkaido wine. This tritium increase may be due to the intense open-air nuclear testing conducted in the French Polynesia Islands (Tadros et al., 2014). The Tokyo record was then scaled by a factor 2.1 to account for the higher tritium concentrations at the higher latitude location (black curve) of Hokkaido. Two pronounced spikes in rain and Hokkaido wine suggest that the tritium record from wine is delayed by approximately one year; this can be attributed to a time delay of recharge of shallow young groundwater. Therefore, the tritium input was shifted and decay corrected by one year. The resulting input curve aligns with the Kofu wine record and overlaps the Tokyo and NIRS tritium records, see Fig. 5. From year 2007, monthly tritium values in precipitation measured by JCAC demonstrate a declining trend with a small tritium spike in March 2011 due to the Fukushima accident tritium release (Matsumoto et al., 2013). This indicates that the JCAC record of the Tokyo area is relatively un-impacted by local tritium sources at present and may be used as the master record for scaling to other Japanese locations with local data as demonstrated in our approach for the Hokkaido area.

### 4.2 Tritium and stable isotope results

The tritium and stable isotope (D and $^{18}$O) results as well as water chemistry analysis of Hokkaido water samples are summarized in Table 2. The tritium values in June ranged between 4.07 ($\pm$0.07) TU at Tougeshita (#12) and 5.29 ($\pm$0.09) TU at Okukatsura (#7), see locations of sampling points in Fig. 3. Otarunai (#1a) had a tritium concentration of 4.26 ($\pm$0.07) TU similar to Piukenai (#11) with 4.37 ($\pm$0.07) TU and Ikutora (#8a) with 4.66 ($\pm$0.07) TU. Rubeshinai (#9a) had a tritium concentration of 4.91 ($\pm$0.07) TU similar to Okukatsura (#7). The high tritium values of Okukatsura and Rubeshinai may be explained by contributions of snowmelt water during the time of sampling at baseflow (Fig. 4). In June, we did not analyse the tritium concentrations of the snow pack because we have estimated the tritium concentration of the infiltrating groundwater from the long-term records of rain data in Japan, and from tritium in Hokkaido wine which during growth utilised young infiltrated groundwater. However, we emphasize that tritium measurements in precipitation are essential for local tritium studies. These tritium precipitation measurements provide the site-specific information for scaling of the established input function to nearby locations. For the Hokkaido area, we have started collection of precipitation and snow core samples for tritium analysis from the January-April 2016 winter season at several sites of the Ishikari River basin. This information will be used to fine-tune the local tritium input within the various Hokkaido sub-catchments. Construction and

scaling of the long-term time-series tritium input function using local data will be included in a separate publication on the tritium input in Japanese precipitation.

Only one river water sample was analysed in July due to a large rain event that occurred in the Ishikari River basin during the sampling, this had a tritium value of 5.06 (±0.09) TU. A rain sample was collected at Kogen hot spring area, which is upstream of Ishikaridaira (#10) station, on July 26th 2014 and had a tritium concentration of 9.16 (±0.14) TU. In October, the river water tritium concentrations were slightly lower than the summer values except for the Piukenai station. The Otarunai (#1b), Rubeshinai (#9b), and Ikutora (#8b) had tritium concentrations of 4.18 (±0.06) TU, 4.82 (±0.07) TU, and 4.45 (±0.07) TU, respectively. For the Takinosawa (#2), the tritium concentration was 4.11 (±0.06) TU and is similar to that of Otarunai (#1b), which is located in the neighbouring river catchment with similar hydrogeology. This result suggests that the two river catchments have similar groundwater dynamics and could be draining one subsurface groundwater storage. This is also indicated by similarity of silica concentrations (Table 2), while higher concentrations of calcium and magnesium of Takinosawa (#2) are an indication of different geological materials such as non-alkaline felsic volcanic rocks (see Fig. 3). A similar situation could be occurring in the other neighbouring river catchments such as Izariirisawa (#3) and Honryujyoryu (#4), which had similar tritium concentrations of 3.83 (±0.07) TU and 3.93 (±0.06) TU, respectively. However, neighbouring river catchments Kouryu (#5) and Hakusen (#6) may have different groundwater dynamics as indicated by tritium as well as calcium and sulphate concentrations (Table 2). The lowest tritium concentration of 3.75 (±0.07) TU was analyzed at Kouryu (#5), which is similar to Izariirisawa (#3) and Honryujyoryu (#4), while Hakusen (#6) tritium of 4.10 (±0.06) TU is similar to Otarunai (#1) and Takinosawa (#2) results. For other samples, the Ishikaridaira (#10b) with a tritium value of 4.85 (±0.07) TU is located next to Rubeshinai (#9b) with 4.82 (±0.07) TU while having slightly different calcium and silica concentrations.

The relationship between tritium and δD is shown in Fig. 6a, and that for δD and $\delta^{18}$O in Fig. 6b. Figure 6b shows that most of the river stable isotope data plot near a local meteoric water line with an intercept of 19, while the one rain sample plots closer to the global meteoric water line. No significant relationship was observed between analysed tritium and water chemistry in Table 2. Samples collected at low elevations (#12, #5, and #6) had the lowest concentrations of tritium, and the most positive δD and $\delta^{18}$O values. Although the Izariirisawa (#3) and Honryujyoryu (#4) samples were collected at 490 masl and had similar values of tritium, the δD and $\delta^{18}$O values of Honryujyoryu are much lower than those of Izariirisawa. Otarunai (#1b) and Takinosawa (#2) samples collected in the same area at about 430 masl had similar values of tritium, δD and $\delta^{18}$O to the Piukenai (#11) June sample. This discrepancy between June and October values may be attributed to the snowmelt water contribution that was occurring during the June sampling trip. The Rubeshinai (#9a-b) and Ishikaridaira (#10a-b) samples, which were collected at about 845 masl, have similar tritium, δD and $\delta^{18}$O values to the Okukatsura (#7) sample collected at 190 masl. These results indicate that the tritium values in coastal rain may be diluted by freshly evaporated ocean water. Therefore, the tritium input of the coastal catchments was corrected by 2.5% (equivalent of MTT of c. 0.5 yrs) towards lower values, and for the catchments with a more negative stable isotope signature by 2.5% towards higher values.

## 4.3 Simulated groundwater transit times

Table 3 summarizes the estimated groundwater MTTs from measured tritium river concentrations using several scaling factors, and the exponential(70%)-piston flow(30%) model (E70%PM) (Fig. 7). In Table 3, MTTs are selected from the MTT range between 1 and 100 yrs with relevant scaling factors of tritium input, which was done by automated simulation with a developed Visual Basic module in TracerLPM (Jurgens et al., 2012). The scaling factor of the Hokkaido tritium curve was re-adjusted using the stable isotope composition of each sample and ranges between 2.05 and 2.15 (Table 3). In regards to the choice of the EPM, we selected this model based on the hydrogeological similarity of Hokkaido to New Zealand settings from Morgenstern et al. (2010). Morgenstern et al. (2010) found that the piston flow component purely due to flow through the unsaturated zone in the headwater catchment is greater than 20%. An exponential-piston flow model with 30% contribution of piston flow within the total flow volume therefore seems appropriate in this study. In Fig 7a, we demonstrate several groundwater transit time distributions for estimated MTTs with E70%PM in Table 3. Each cumulative distribution function describes the proportion of water with transit times up to the specific transit time and MTTs are at about 0.63 of total flow volume (Fig. 7a). The horizontal initial parts of these transit time distributions on the x-axis represent the 30% piston flow sections and range from 0.3 to 7 yrs, see Fig. 7a. In Table 3, the good correspondence between analyzed and simulated tritium values is demonstrated by small relative error values, which is equivalent to one sigma error of tritium analysis with values of about 1.5%. These relative errors are plotted between 1 and 100 yrs for selected unique and non-unique cases in Fig 7b. The serrated pattern of relative errors is transferred from simulated tritium concentrations and is due to the monthly time step in tritium input of the TracerLPM. The smallest relative errors of Otarunai (#1a and #1b) and Kouryu (#5) demonstrate one MTT solution estimated with E70%PM (Fig. 7b). The similar pattern is reported for six study catchments in Table 3. Despite either the youngest MTTs of c. 0.1 yrs or the oldest MTTs above 100 yrs being excluded as improbable, we find several equally good fits for some stations indicating that the MTT solution is non-unique (i.e. water with different MTTs can have similar tritium concentrations) (Table 3). For example, we have two solutions for groundwater transit times at Ikutora (#8a-b), Rubeshinai (#9a-b), Ishikaridaira (#10a-b), Piukenai (#11) and Tougeshita (#12) while Okukatsura (#7) has three solutions: very young (e.g., Okukatsura MTT=1 yrs), young (e.g., Okukatsura MTT=4 yrs), and old (MTT=23 yrs). This is due to the interference by the bomb-tritium that is still present in Hokkaido groundwater and will take a number of years to decay and flush out. Having tritium-series measurements with 3 year intervals will enable us to choose either the young or old MTT value and therefore to reduce the ambiguity of the simulated transit times.

To illustrate this point, historical and future tritium concentrations at baseflow are demonstrated for the full range of MTTs in Fig. 8, where tritium concentrations at baseflow are simulated with E70%PM. This model used the Hokkaido recharge from 1950 to 2015 established here and forecasted monthly long-term average tritium values from 2015 to 2030 with the assumption of stable tritium concentrations in rain similar to those of the last five years. From the simulated tritium concentrations, tritium in river water will reach levels similar to those analysed in the Southern Hemisphere in the next decade as also demonstrated by Stewart and Morgenstern (2016). This implies that one tritium river water sample may then

be sufficient to estimate unique groundwater MTTs and therefore robust storage volumes for most of the Japanese catchments, if assumptions about transit time distributions are made. Despite the present ambiguity, we can attempt to utilize river water chemistry in Table 2 for selecting young or old MTT and estimating groundwater storage volume. For these locations with non-unique MTTs, we evaluate the change of chemical composition between two sampling dates as well as assuming an increase of silica and other ion concentrations with MTT (Morgenstern et al., 2010, 2015). For the locations with one collected sample, the lowest silica concentrations are observed for Okukatsura (#7) with 3.34 mg $L^{-1}$ and Tougeshita (#12) with 5.46 mg $L^{-1}$, compared to the other collected samples of Hokkaido study catchments, indicating higher likelihood of the younger MTTs or differences in dissolution rates. From this assumption, we select MTTs of 1 and 4 yrs for Okukatsura (#7) and of 11 yrs for Tougeshita (#12). Following the same pattern, October samples of Ikutora (#8b) and Rubeshinai (#9b) have slightly higher ion concentrations including silica compared to June samples while demonstrating decrease in analysed tritium concentrations. This may indicate older MTTs in October leading to MTTs of 17 yrs for Ikutora (#8b) and 20 yrs for Rubeshinai (#9b).

In our tritium interpretation, we also found only one MTT solution of groundwater transit times for seven river samples in six catchments (Table 3): Otarunai (#1a-b), Takinosawa (#2), Izariirisawa (#3), Honryujyoryu (#4), Koryu (#5), and Hakusen (#6). To validate this finding, we sampled river water near Otarunai station on February 24[th], 2016, to investigate tritium concentrations in winter baseflow conditions at the Sapporo area of Hokkaido. If this river sample gives the same MTTs, it will confirm that one tritium sample is sufficient to estimate unique MTT in these and possibly other Japanese river catchments. Moreover, the result of similar tritium concentrations and MTTs for the neighbouring river catchments indicates similar groundwater flow and drainage patterns, which are controlled by hydrogeological settings. These six river catchments are situated in similar Quaternary lavas and Tertiary propylite formations (Fig. 3) while having different river catchment features such as mean annual flows, drainage areas, terrain slopes, etc. (Table 1). The chemical concentrations may also support this hypothesis of similar groundwater flow and drainage patterns in these catchments. For example, Mg/Ca ion ratio estimated from meq $L^{-1}$ concentrations is about 0.6 for Otarunai (#1), Takinosawa (#2) and Hakusen (#6) river catchments and is about 0.43 for Izariirisawa (#3), Honryujyoryu (#4), and Koryu (#5) river catchments. It may also be possible that the neighbouring river catchments have only one subsurface groundwater storage supporting river baseflows at different river catchments. It is known that the groundwater systems can have different boundaries than river catchments and one subsurface groundwater storage can be drained by neighbouring river catchments (Grannemann et al., 2000; Gusyev et al., 2014). In that case, a subsurface groundwater storage shared by the Otarunai (#1) and Takinosawa (#2) river catchments contributes inflow to the Jyozankei dam and regional groundwater recharge to Sapporo alluvial aquifer (Dim et al., 2002) indicating important implications for water availability for dam inflows and groundwater abstraction at Sapporo city (Sakata and Ikeda, 2013). However, a detailed hydrogeologic study is required to further investigate this hypothesis.

Vulnerability of stable isotope-based MTT to aggregation error has been recently discussed by Kirchner (2016a, b). Kirchner (2016a) demonstrated the MTT aggregation error of [18]O using hypothetical transit times at two neighbouring headwater catchments and indicated a need for similar evaluation for tritium-inferred ages. It seems that tritium and MTT data of our

neighbouring catchments in Hokkaido can be used to evaluate the MTT aggregation error between real and apparent MTTs. For this evaluation, we select Otarunai (#1) with an area of 68 km$^2$ and Takinosawa (#2) with an area of 44 km$^2$, which are situated in similar hydrogeological settings in neighbouring catchments (Figs. 2 and 3). On October 24$^{th}$, Otarunai (#1b) had tritium of 4.18 TU at baseflow of 3.66 m$^3$ s$^{-1}$ and Takinosawa (#2) 4.11 TU at 0.53 m$^3$ s$^{-1}$. The simulated MTTs with

E70%PM are 14 and 13 yrs for Otarunai (#1b) and Takinosawa (#2), respectively (Table 3). The combined discharge for these two locations is 4.19 m$^3$ s$^{-1}$ leading to the tritium concentration of 4.12 TU and MTT of 13.9 yrs. From this tritium concentration of 4.12 TU, we use E70%PM with the same scaling factor of 2.1 to simulate an apparent MTT of 13.6 yrs, which is very close to the combined MTT of 13.9 yrs. As a result, the baseflow MTT of these two catchments has very low MTT aggregation error (about 2%) demonstrating a good match between the two MTTs. It is also important to note that the

apparent MTT of 13.6 yrs remains unique (i.e. it is the only best-fit solution in the range of MTTs between 1 and 100 years). This point is illustrated in the inset in Figure 8 with the unique MTT solution shown in blue and non-unique solutions shown in red (the detailed discussion is provided by Stewart and Morgenstern (2016)). From this example, we find that neighbouring catchments with topographic heterogeneity have low MTT aggregation error under the following conditions: 1) similar MTTs and tritium concentrations at baseflow; 2) unique MTT solutions (no interference of bomb-peak tritium), and

3) similar transit time distributions of groundwater flow (due to hydrogeologic similarity). Once these conditions are violated, the MTT aggregation error of neighbouring catchments may be significant. This preliminary finding should be further investigated for other tritium cases in light of the discussion by Kirchner (2016a, b).

## 4.4 Simulated groundwater storage and saturated thickness

We estimate ranges of groundwater storage volumes between 0.013 and 5.07 km$^3$ and saturated water thicknesses between

0.2 and 25 m by using Equation 3. These values of saturated water thickness are smaller than the recent estimates of groundwater storage thickness of 180 m by Gleeson et al. (2016) and much larger than the 0.055 m saturated water thickness of young (MTT of 0.2 yrs) terrestrial water identified by Jasechko et al. (2016). For the Otarunai (#1) and Takinosawa (#2), we used MTTs of 13 and 14 yrs with baseflow values of 3.66 and 0.53 m$^3$ s$^{-1}$ to find groundwater storages of 1.62 and 0.22 km$^3$, respectively. Dividing these two volumes by the respective drainage areas of 64 and 14 km$^2$ (Table 1) we find saturated

thicknesses of water of 25.3 m for Otarunai and 15.7 m for Takinosawa. For nearby catchments the saturated water thickness of the Izariirisawa (#3) with catchment area of 42 km$^2$ is 6.9 m (estimated from 0.29 km$^3$ storage based on MTT of 13 yrs and 0.71 m$^3$ s$^{-1}$ baseflow). The Honryujyuryu (#4) has 14.6 m saturated water thickness (estimated from 0.95 km$^3$ storage obtained at 2.3 m$^3$ s$^{-1}$ baseflow and catchment area of 65 km$^2$). The Ikutora (#8) has the largest drainage area of 377 km$^2$ and saturated water thickness of 13.4 m (estimated from 5.07 km$^3$ storage using MTT of 17 yrs at 9.5 m$^3$ s$^{-1}$ baseflow). The

Rubeshinai (#9) has 7.3 m saturated thickness of water (estimated from 0.33 km$^3$ storage using MTT of 20 yrs at 0.53 m$^3$ s$^{-1}$ baseflow and catchment area of 45 km$^2$), while the saturated water thickness of Ishikaridaira (#10) is about 24 m (estimated from 2.72 km$^3$ storage obtained using MTT of 22 yrs at 3.92 m$^3$ s$^{-1}$ baseflow and catchment area of 113 km$^2$). The Tougeshita (#12) has the saturated thickness of water of 1.8 m (from catchment area of 49 km$^2$ and 0.094 km$^3$ of storage

with MTT=11 yrs). From Eq. (3) with young MTTs, we estimate groundwater volume of 0.013 km$^3$ with MTT=1 yrs and 0.052 km$^3$ with MTT=4 yrs for Okukatsura (#7) and find the saturated water thickness of 0.2 and 0.9 m with the catchment area of 56 km$^2$ and 0.013 and 0.052 km$^3$ volumes, respectively.

We indicate the importance of groundwater storage characterization with tritium river water samples at baseflow by a comparison of stable isotopes and tritium simulated MTTs. Out of sixteen tritium samples, only three samples have MTTs below 5 years at baseflow while modelled MTTs of 12 samples range between 6 and 23 yrs (Table 3). For these 12 samples, only tritium analysis allows us to characterize groundwater storage with long transit times from years to decades due to the limitation of $^{18}$O and $^2$H stable isotopes for identifying MTTs older than 5 yr (McGuire and McDonnell, 2006). This order-of-magnitude difference in sensitivity between the stable isotope and the tritium methods will naturally result in the stable isotope method being preferably applied to short transit time and low volume systems, and the tritium method to long transit time and large volume systems. Therefore, the difference in stable isotope and tritium-derived water storages is driven by the difference in MTTs. In addition, the aggregation error proposed by Kirchner (2016a, b) may cause stable isotope derived MTTs to underestimate storage. It has been demonstrated that the use of stable isotopes enables MTT simulation in the range of a few months up to about five years (McGuire and McDonnell, 2006) for groundwater storage volume estimates (Małoszewski et al., 1992; Leopoldo et al., 1998; McGuire et al., 2002; Jasechko et al., 2016). Leopoldo et al. (1998) simulated MTTs of about 0.4 years with $^{18}$O values in two Brazilian agricultural watersheds of 1.6 km$^2$ and 3.3 km$^2$ and obtained groundwater volume of 0.0001 km$^3$ with 0.06 m saturated thickness of water for Bufalos watershed and 0.00037 km$^3$ with 0.11 m saturated thickness of water for Paraiso watershed. In cases when simulated MTTs from stable isotopes and tritium have similar values, the groundwater storage volumes do not differ much. For example, Małoszewski et al. (1992) reported similar estimated MTTs of about 4.1 years with $^{18}$O and tritium in the Wimbachtal valley watershed of 33 km$^2$ and computed subsurface water volume of about 0.22 km$^3$ with 6.6 m of saturated thickness of water. MTTs obtained with stable isotope and tritium tracers in many catchments have been summarized by Stewart et al. (2010). Following Kirchner (2016a, b) the vulnerabilities of tritium based MTTs to aggregation error needs to be investigated further.

From these findings, we suggest that the changes of subsurface groundwater storage, which supplies the majority of baseflow especially during winter and dry summer conditions in Hokkaido, need to be accounted for in the management of water resources in our study catchments. The importance of the subsurface groundwater storages is emphasized by comparing them with the normalized storages of the five dams (i.e. water storage in the reservoir divided by the corresponding catchment area) (Table 1). For these five dams, this average saturated thickness of water ranges between 0.1 and 0.8 m and is much smaller than storage in the study headwater catchments, which have saturated thicknesses of water between 0.2 and 25 m. To demonstrate groundwater storage changes, we simulate the hourly change of estimated groundwater volume at Otarunai station from June 2014 to February 2016 (Fig. 9). The tritium sampling times are shown by vertical lines, see Fig. 9. In this simple approach, the lumped numerical model does not include any sophisticated calculations such as energy balance, delay in recharge, soil types, etc., and only simulates the changes of saturated groundwater storage that receives recharge from infiltrated soil water and contributes to the baseflow discharge. In our

simulation, the groundwater storage is recharged by 20% of precipitation and 80% of snow melt water, which is estimated from hourly snow depth using snow water equivalent of 0.4, and is drained by baseflow, which was estimated from hourly river discharge data (Fig. 4a). We obtained these recharge rates from a range of field values reported by Iwata et al. (2010) for the Tokachi site in Hokkaido. Iwata et al. (2010) investigated water infiltration rates at 0.2 and 1.05 m soil depth from

2002 to 2006 and reported that the largest rates of soil water infiltration of between 79% and 85% occurred during the spring snow melt season compared to the summer-fall water infiltration rates of 20-25% in 2002. From Eq. (5) with the estimated volume of about 1.62 km$^3$ we find groundwater volume of 1.65 km$^3$ and saturated water thickness of 25.6 m on June 4$^{th}$ 2014 using Eq. (3). This simulation demonstrates a decline of groundwater volume by 0.03 km$^3$ and of saturated water thickness by 0.3 m from June to October, while having some small spikes during periods of high groundwater recharge in August and

September 2014. From October 24$^{th}$, the groundwater volume declines over December-February reaching the smallest volume of 1.59 km$^3$, which is equivalent to saturated water thickness of 24.8 m. Once the snowmelt season starts in mid-March 2015, the accumulated snow layer of up to 3.1 m melts and snow melt water replenishes groundwater storage until the end of snowmelt season, see Fig. 9. As a result, the groundwater volume of subsurface storage equals 1.64 km$^3$ and saturated water thickness 25.6 m on May 14$^{th}$ 2015. From June, the subsurface storage is again drained by baseflow resulting in 1.63

15   km$^3$ of groundwater volume and 25.4 m of saturated water thickness on June 4$^{th}$ 2015. This difference of groundwater volume between June 4$^{th}$ 2014 and 2015 is due to drier weather conditions in year 2014 with annual precipitation of 0.86 m compared to annual precipitation of 1.00 m in year 2015. The groundwater volume continues to gradually decline due to drainage by baseflow while receiving groundwater recharge from precipitation until October 2015. Once the winter season starts, the groundwater storage is again drained by winter season baseflow of about 1.8 m$^3$s$^{-1}$ reaching 1.58 km$^3$ of

groundwater volume and 24.7 m of saturated water thickness on February 24$^{th}$ 2016 (Fig. 9). After the melting of snow starts in mid-March, the snow melt water of accumulated snow layer recharges the subsurface storage and the groundwater volume is again replenished (not shown). This result indicates two important points: 1) the role of snow hydrology in groundwater dynamics demonstrating the impact of a dry winter with little snow on the drought conditions in Hokkaido, and 2) the large groundwater volumes of subsurface storage in the Hokkaido headwater catchments potentially available to maintain

baseflow during prolonged droughts.

## 5 Concluding Remarks

We demonstrated the application of tritium by estimating the groundwater mean transit times (MTTs) and subsurface volumes in headwater catchments of Hokkaido, Japan, from tritium data of river water and precipitation. Sixteen river water samples in Hokkaido were collected in June, July and October 2014 at twelve locations. These locations drain areas between

45 and 377 km$^2$ and are all located upstream of MLIT dams, except Tougeshita station. The collected water samples were analyzed by the Tritium Laboratory, New Zealand, and resulting tritium concentrations ranged between 4.07 TU (±0.07) and

5.29 TU (±0.09) in June and 3.75 TU (±0.07) and 4.85 TU (±0.08) in October 2014. One river sample had 5.09 (±0.09) TU and one rain had 9.16 (±0.14) TU in July 2014.

The tritium record in precipitation was reconstructed from GNIP stations for the Tokyo area and scaled to the Hokkaido area using local data. To estimate MTTs we applied the exponential(70%)-piston flow(30%) model (E70%PM) to the reconstructed tritium record for Hokkaido and obtained non-unique fits of very young, young and old groundwater transit times due to the interference by bomb-peak tritium that is still present in Japanese waters. Having tritium-series measurements with 3 year intervals would enable us to choose either the young or old MTT value and therefore to reduce the ambiguity of the simulated transit times. Eventually, tritium in groundwater will reach natural levels and one tritium river water sample will be sufficient to estimate a robust groundwater storage volume as well as saturated thickness of water in the subsurface. However, we also simulated unique MTT values in six river catchments located near Sapporo city assuming that the system response function (E70%PM) describes catchment flow conditions there. This finding led to two important conclusions: 1) that one tritium sample is sufficient to estimate MTT for most of our watersheds, and 2) that the similar tritium and MTTs of baseflow in adjacent river catchments are controlled by hydrogeological settings resulting in similar groundwater flow and drainage patterns. The unique MTT shown by some of the river watersheds allows us to estimate unambiguous groundwater storage volumes as demonstrated for the Otarunai catchment. For example, the groundwater storage ranges between 0.013 and 5.07 $km^3$ with saturated water thickness from 0.2 and 25 m. Knowledge of groundwater storage volume enables us to investigate changes of groundwater volumes with time and provide useful information for the improvement of numerical models and water resources management especially during droughts. As a result, the adopted approach may be a cost-effective method of characterizing groundwater transit times and volumes of subsurface storage and could be used to improve simulated groundwater dynamics by rainfall-runoff models in future studies.

**Acknowledgements**

We thank offices of Jyozankei, Houheikyo, Kanayama, Katsurazawa, Taisetsu, Chubetsu, and Izarigawa Dams for their support in collecting river water samples and Assistant Prof. T. Oda, University of Tokyo, for conducting chemistry analysis of water samples. We are grateful to Assoc. Prof. T. Hayashi, Akita University, for providing information of tritium in Japanese precipitation, to Dr. N. Nagumo for providing geological information, and Ms. M. Yamamoto for her support of this study.

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

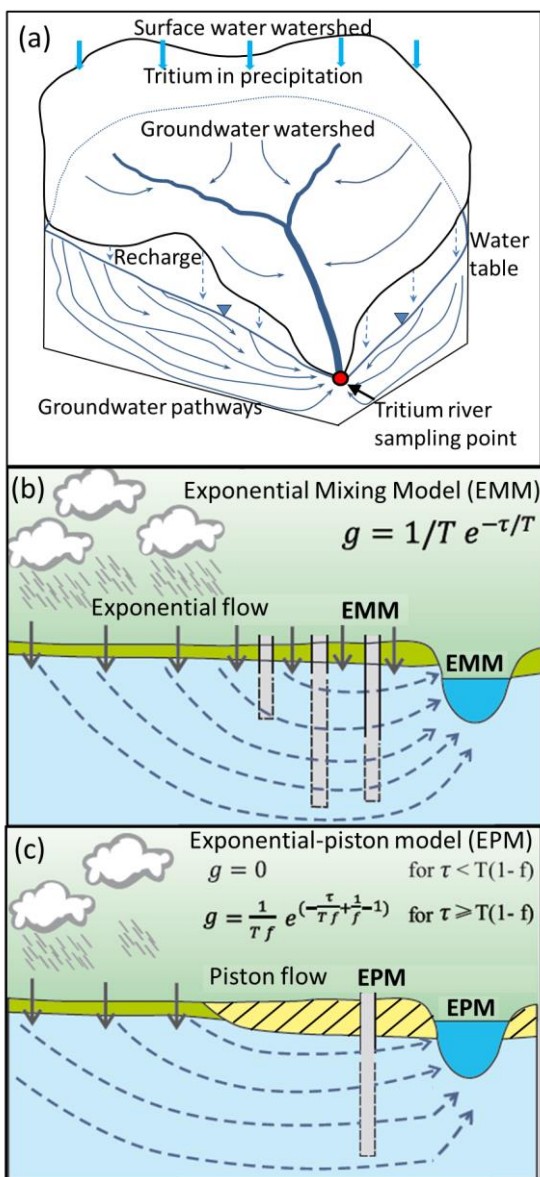

**Figure 1: Conceptual diagram of the tritium cycle in a river and subsurface groundwater storage (a) as the tritium input in precipitation is transformed to the tritium output in river water by passing through the subsurface. These complex dynamics are represented by the exponential mixing model (EMM) for the unconfined aquifer (b) and the exponential-piston model (EPM) for the partially confined aquifer (c).**

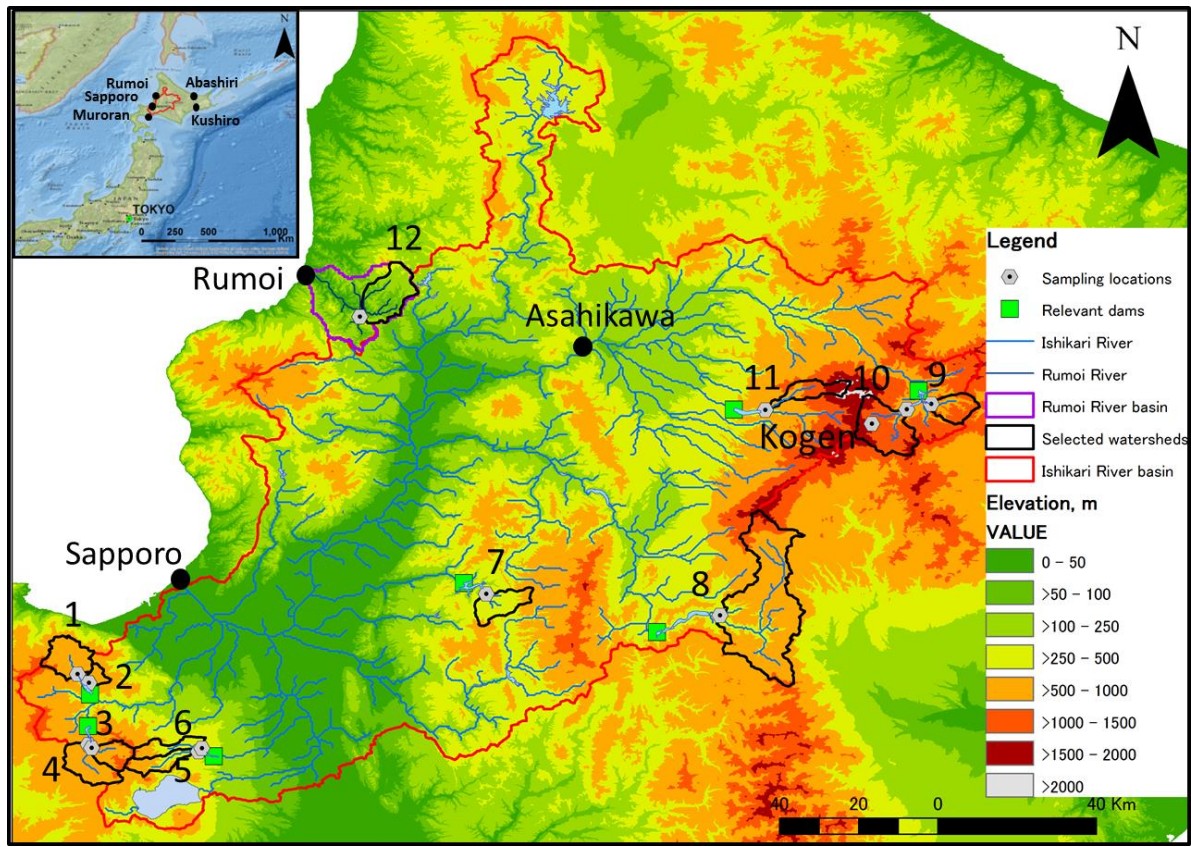

**Figure 2: Location of the Hokkaido study area with the sampling points in the selected watersheds shown by circles.**

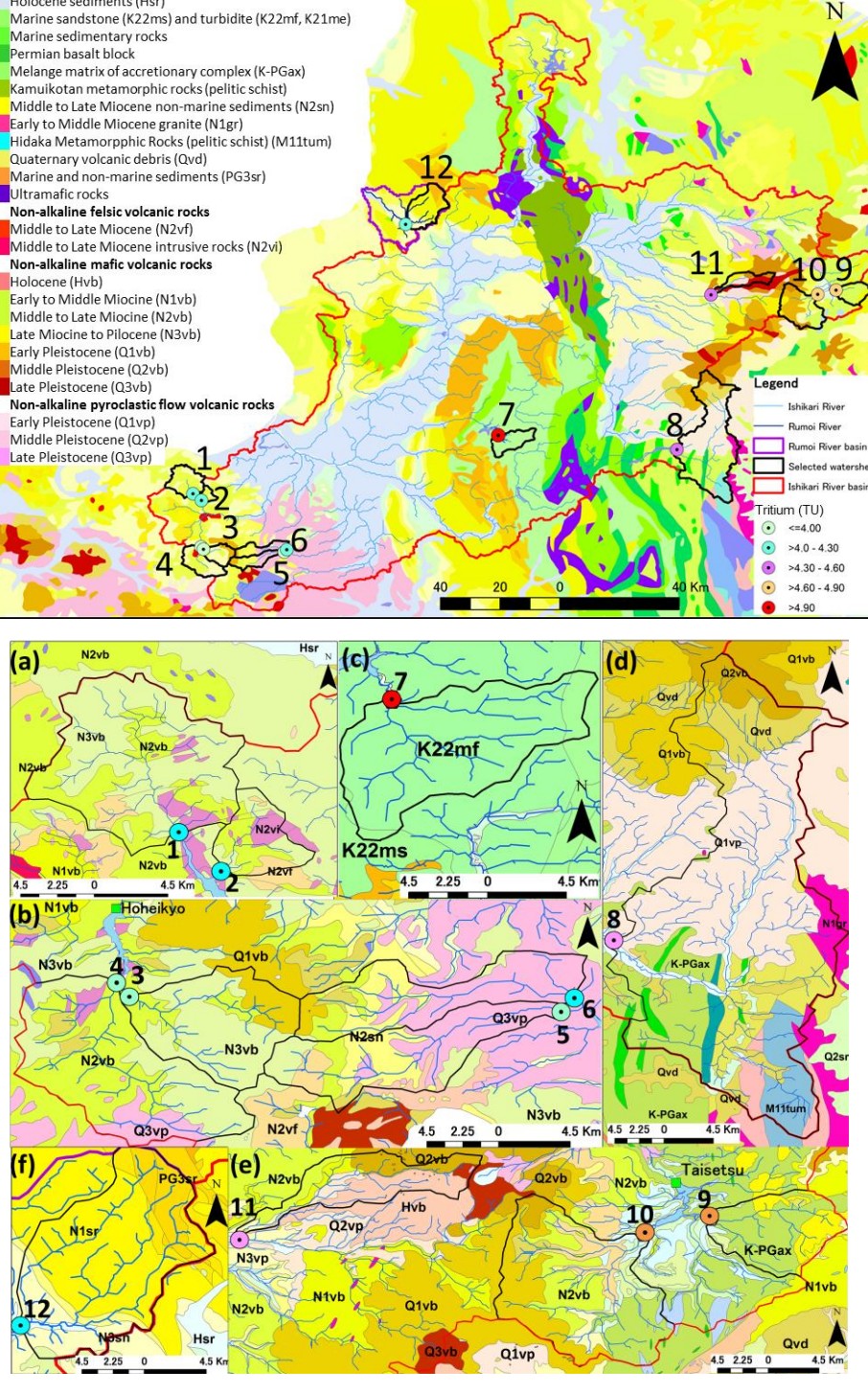

**Figure 3: Geology of the study area from AIST (2012), with zoom-ins on the twelve study watersheds (a-f). Analysed tritium concentrations are demonstrated in colour code.**

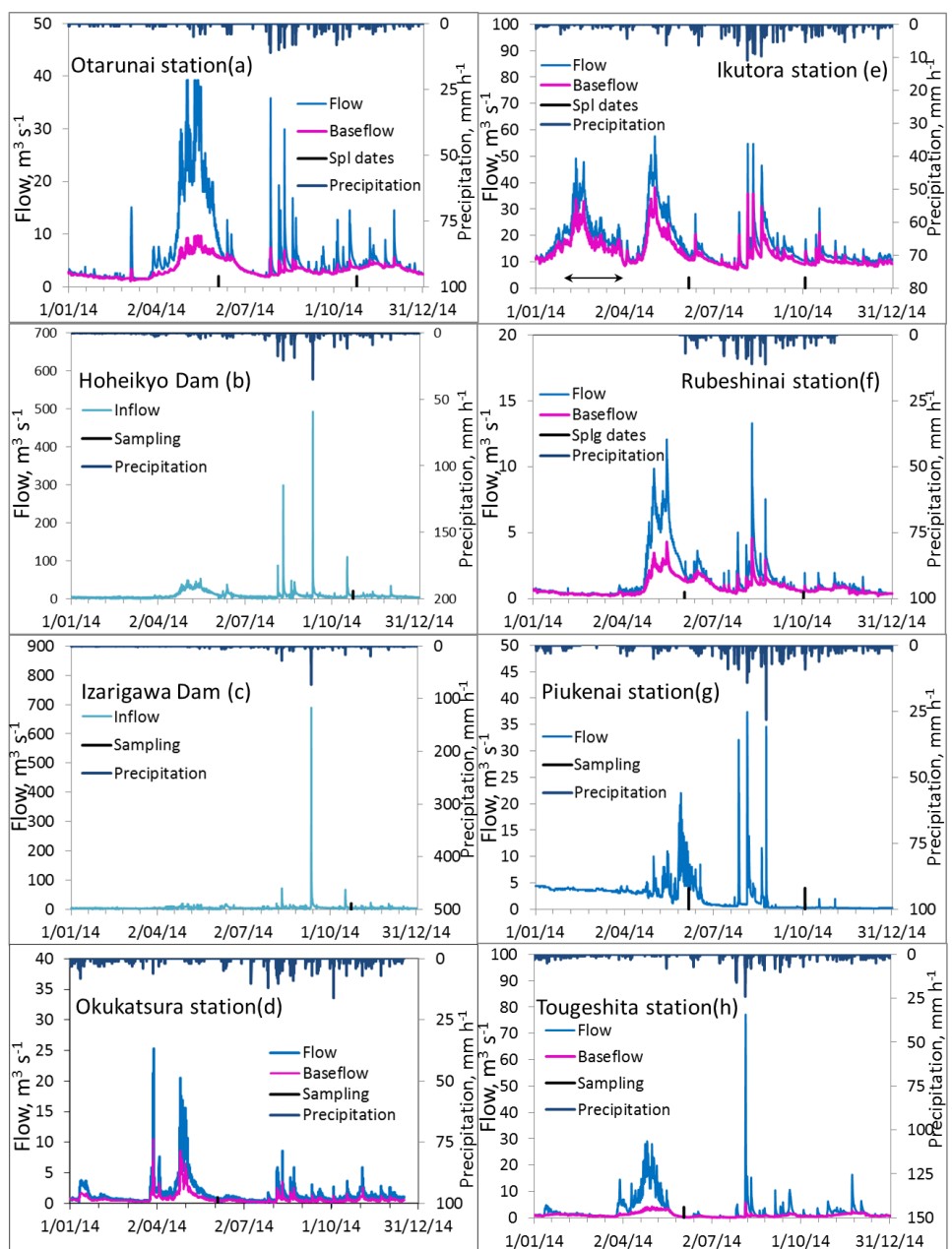

**Figure 4: Hourly river flows at Otarunai station (a), Hoheikyo(c) and Izarigawa (b) dams, Okukatsura station (d), Ikutora station (e), Rubeshinai station (f), Piukenai station (g), and Tougeshita station (h). The sampling times of June, July and October are demonstrated by vertical lines.**

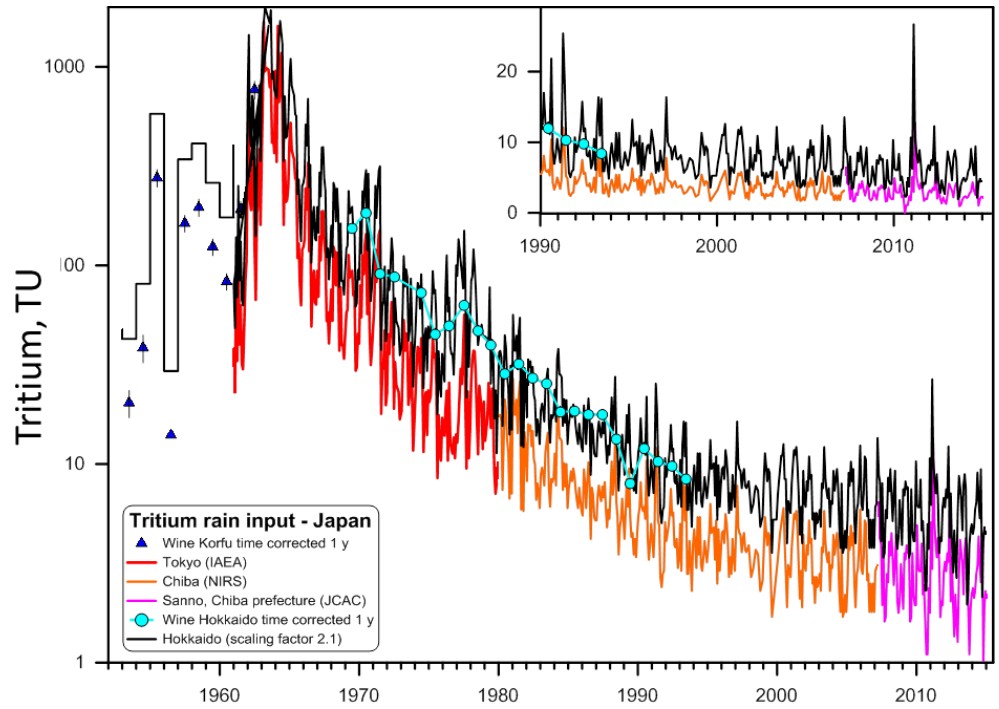

**Figure 5: The constructed tritium time-series in Tokyo precipitation and Hokkaido groundwater recharge using wine data. The precipitation input curve is constructed using tritium data of Kofu wine (1952-1960), IAEA Tokyo station (1961-1975), Chiba NIRS (1976-2007) and Chiba JCAC (2008–to present). The inset shows tritium time-series from 1990 to present.**

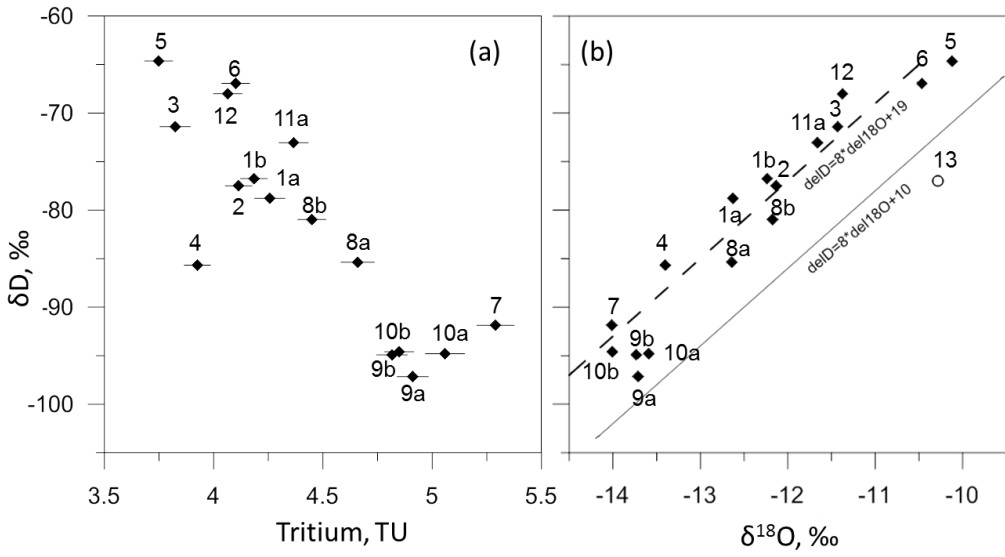

**Figure 6: Relationships between a) δD and tritium (a), and δD and δ[18]O (b) in Hokkaido river waters (diamonds) and rain (open circle). Labels refer to IDs in Table 2.**

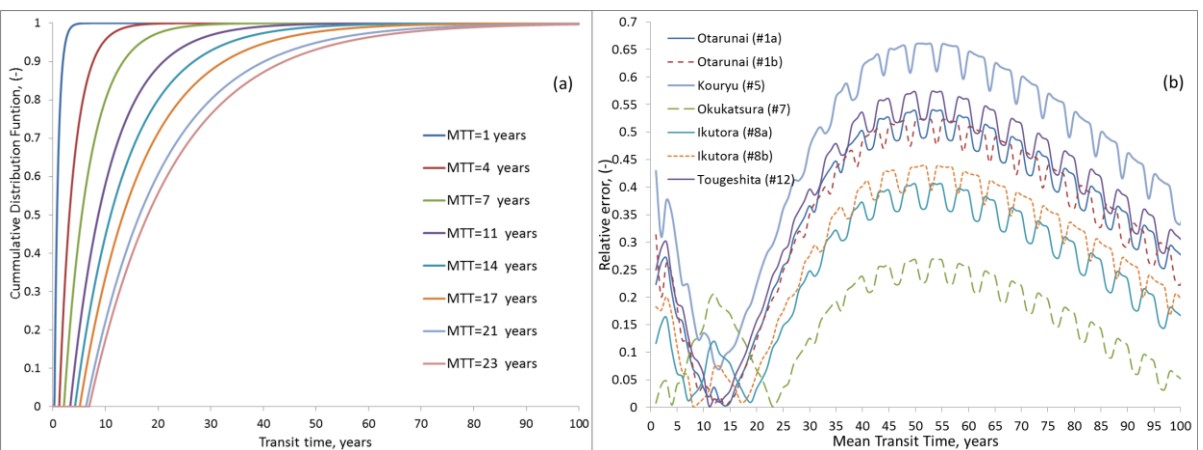

**Figure 7: Transit time distributions of the exponential (70%)-piston flow (30%) model (E70%PM) for obtained MTT solutions (a) and relative error between simulated and analysed tritium at selected locations for MTTs between 1 and 100 yrs (b).**

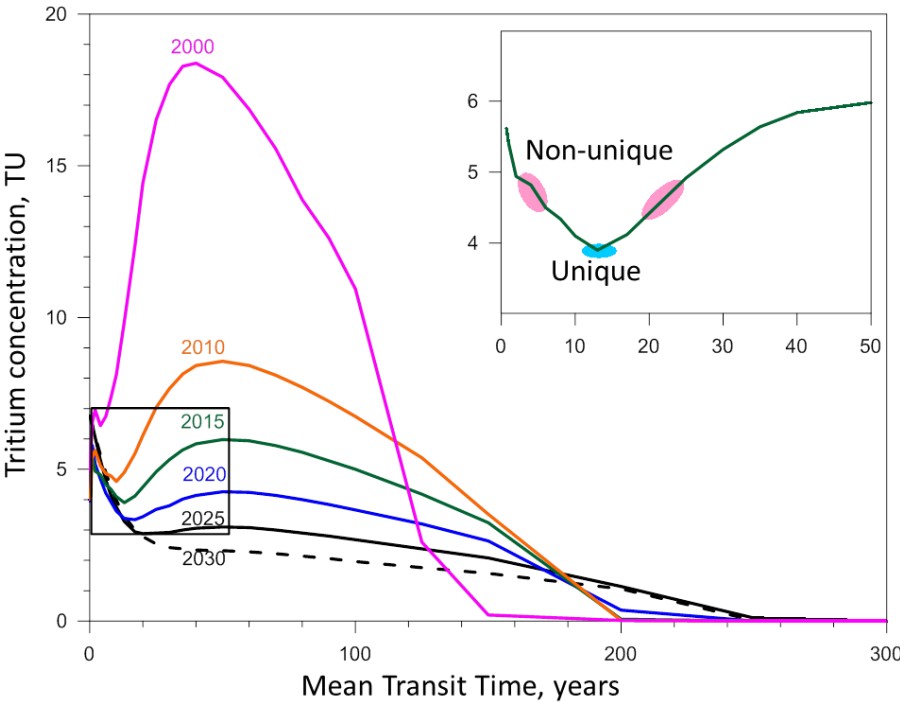

5 **Figure 8: Simulated tritium concentrations in Hokkaido river water for the years 2000, 2010, 2015, 2020, 2025 and 2030 versus groundwater MTT. A factor of 2.0 was used to scale the Tokyo input and EPM with 70% exponential. The inset shows 2015 tritium concentrations with unique MTT indicated in blue and non-unique MTTs (two possibilities) indicated in red.**

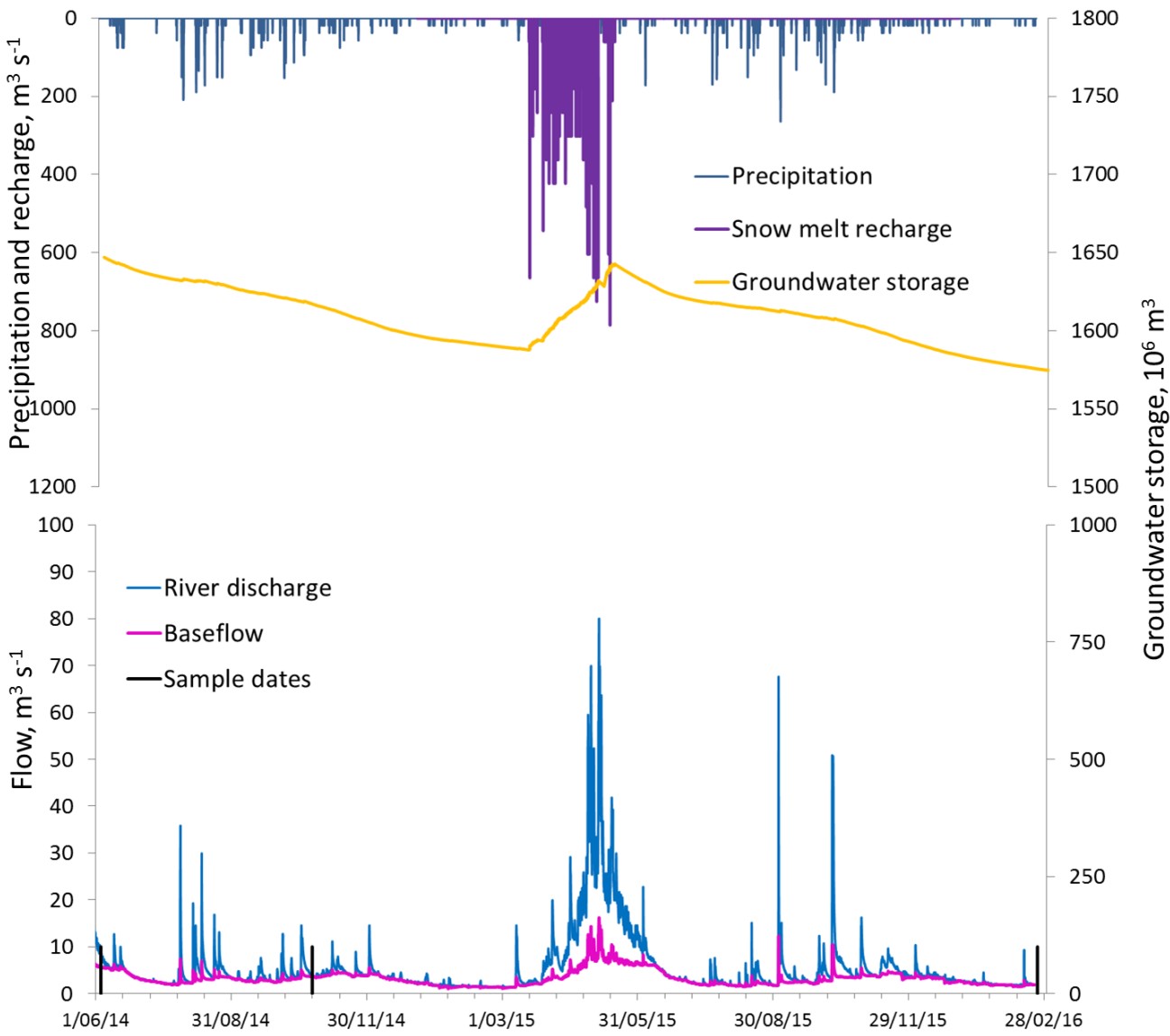

**Figure 9: The change of Otarunai groundwater storage estimated using tritium groundwater volume at baseflow. The groundwater storage is recharged by precipitation and snowmelt and is drained by the baseflow component of river discharge.**

**Table 1. Characteristics of twelve investigated headwater catchments in Hokkaido, Japan.**

| ID | Location | Mean flow, $m^3 s^{-1}$ | Catchment area, $km^2$ | Drainage density, $km^{-1}$ | Mean slope, - | Elevation, masl Point | Mean | Highest | Surface and subsurface geology | Name of Dam | Capacity, $km^3$ | Catchment area, $km^2$ |
|----|----------|------|------|------|------|------|------|------|--------------------------------|-------------|--------|--------|
| 1 | Otarunai | 5.34 | 68 | 1.4 | 0.25 | 397 | 736 | 1278 | Propylite, Lavas (augite hypersthene andesite) | Jyozankei | 0.082 | 104 |
| 2 | Takinosawa | 0.76 | 14 | 1.0 | 0.31 | 398 | 696 | 1061 | Propylite, Lavas (augite hypersthene andesite), intrusive rocks | | | |
| 3 | Izariirisawa | 3.30 | 42 | 1.2 | 0.26 | 496 | 905 | 1304 | Lavas (augite hypersthene andesite), propylite, quartz | Hoheikyo | 0.043 | 159 |
| 4 | Honryujyoryu | 2.93 | 65 | 1.4 | 0.19 | 484 | 788 | 1211 | Lavas (hypersthene andesite), propylite, quartz | | | |
| 5 | Kouryu | 2.43 | 40 | 1.6 | 0.25 | 187 | 567 | 1297 | Lavas (augite hypersthene andesite), propylite, shale | Izarigawa | 0.015 | 113 |
| 6 | Hakusen | 1.95 | 42 | 1.5 | 0.20 | 202 | 506 | 1176 | Lavas (augite hypersthene andesite), propylite, sandstone | | | |
| 7 | Okukatsura | 2.72 | 56 | 1.4 | 0.24 | 187 | 419 | 1012 | Sandy siltstone, siltstone, sandstone | Katsurazawa | 0.093 | 299 |
| 8 | Ikutora | 22.59 | 377 | 1.6 | 0.19 | 353 | 706 | 1865 | Rhyolitic welded tuff, lavas, metamorphic and igneous rocks | Kanayama | 0.151 | 470 |
| 9 | Rubeshinai | 1.49 | 45 | 1.3 | 0.27 | 808 | 1034 | 1364 | Lavas, sandstone and slate, conglomerate | Taisetsu | 0.093 | 234 |
| 10 | Ishikaridaira | 9.21 | 113 | 1.5 | 0.26 | 831 | 1342 | 2163 | Lavas (hypersthene augite andesite), slate | | | |
| 11 | Piukenai | 2.66 | 60 | 1.4 | 0.21 | 431 | 1252 | 2247 | Lavas (hornblende hypersthene augite andesite), welded tuff | Chubetsu | 0.066 | 292 |
| 12 | Tougeshita | 2.34 | 49 | 2.5 | 0.16 | 22 | 127 | 712 | Sandstone & sandy alternation of sandstone & mudstone | Rumoi | 0.023 | 42 |

**Table 2. Tritium, stable isotope and chemistry results for the Hokkaido river and rain water samples.**

| ID | Location | Date | Time | Flow, $m^3 s^{-1}$ | $^3H$, TU | ±δ, TU | dD, ‰ | ±dD, ‰ | $d^{18}O$, ‰ | $±d^{18}O$, ‰ | Cl, $mg L^{-1}$ | NO3-N, $mg L^{-1}$ | SO4-S, $mg L^{-1}$ | Na, $mg L^{-1}$ | NH4-N, $mg L^{-1}$ | K, $mg L^{-1}$ | Mg, $mg L^{-1}$ | Ca, $mg L^{-1}$ | Si, $mg L^{-1}$ |
|---|---|---|---|---|---|---|---|---|---|---|---|---|---|---|---|---|---|---|---|
| 1a | Otarunai | 2014/06/04 | 10:55 | 8.26 | 4.257 | 0.070 | -78.78 | 2.27 | -12.63 | 0.48 | 4.36 | 0.23 | 1.37 | 3.67 | 0.00 | 0.45 | 1.18 | 3.50 | 6.16 |
| 1b | Otarunai | 2014/10/24 | 10:20 | 3.82 | 4.184 | 0.063 | -76.75 | 1.6 | -12.24 | 0.44 | 5.46 | 0.25 | 2.16 | 4.63 | 0.00 | 0.54 | 1.80 | 4.76 | 7.12 |
| 2 | Takinosawa | 2014/10/24 | 11:00 | 0.53 | 4.114 | 0.062 | -77.50 | 1.45 | -12.13 | 0.45 | 4.61 | 0.28 | 5.02 | 4.91 | 0.00 | 0.56 | 3.04 | 8.34 | 7.09 |
| 3 | Izariirisawa | 2014/10/23 | - | 0.71 | 3.825 | 0.070 | -71.41 | 1.26 | -11.43 | 0.45 | 3.30 | 0.25 | 4.81 | 4.03 | 0.00 | 0.52 | 2.23 | 8.44 | 7.71 |
| 4 | Honryujyuryu | 2014/10/23 | - | 2.31 | 3.926 | 0.061 | -85.67 | 1.12 | -13.40 | 0.43 | 3.91 | 0.17 | 3.49 | 3.23 | 0.00 | 0.45 | 1.51 | 5.37 | 6.53 |
| 5 | Kouryu | 2014/10/22 | 13:24 | - | 3.748 | 0.065 | -64.65 | 1.56 | -10.12 | 0.40 | 4.13 | 0.24 | 5.79 | 4.91 | 0.00 | 1.12 | 2.32 | 9.62 | 13.52 |
| 6 | Hakusen | 2014/10/22 | 15:45 | - | 4.101 | 0.064 | -66.94 | 1.91 | -10.47 | 0.44 | 4.58 | 0.24 | 3.76 | 4.94 | 0.00 | 1.21 | 2.04 | 5.74 | 15.14 |
| 7 | Okukatsura | 2014/06/04 | 17:30 | 0.48 | 5.290 | 0.086 | -91.87 | 0.84 | -14.01 | 0.43 | 3.25 | 0.00 | 6.09 | 6.74 | 0.00 | 1.23 | 1.74 | 15.79 | 3.34 |
| 8a | Ikutora | 2014/06/06 | 14:30 | 12.35 | 4.659 | 0.077 | -85.37 | 0.84 | -12.64 | 0.40 | 1.86 | 0.25 | 2.36 | 3.21 | 0.00 | 1.04 | 1.13 | 5.39 | 9.27 |
| 8b | Ikutora | 2014/10/03 | 11:00 | 10.99 | 4.449 | 0.065 | -80.97 | 1.29 | -12.17 | 0.47 | 2.13 | 0.30 | 2.47 | 3.63 | 0.00 | 1.18 | 1.36 | 6.12 | 9.46 |
| 9a | Rubeshinai | 2014/06/05 | 17:03 | 1.66 | 4.911 | 0.072 | -97.15 | 1.66 | -13.71 | 0.43 | 1.88 | 0.00 | 1.96 | 3.15 | 0.00 | 0.68 | 1.66 | 7.14 | 7.34 |
| 9b | Rubeshinai | 2014/10/02 | 14:20 | 0.53 | 4.816 | 0.071 | -94.91 | 0.82 | -13.73 | 0.41 | 2.23 | 0.18 | 2.11 | 3.58 | 0.00 | 0.74 | 2.01 | 8.57 | 8.13 |
| 10a | Ishikaridaira | 2014/07/26 | 12:45 | 6.42 | 5.059 | 0.090 | -94.80 | 1.14 | -13.59 | 0.40 | 1.93 | 0.24 | 2.09 | 3.69 | 0.00 | 0.74 | 1.92 | 8.33 | 8.62 |
| 10b | Ishikaridaira | 2014/10/02 | 15:00 | 3.92 | 4.849 | 0.068 | -94.59 | 1.32 | -14.01 | 0.47 | 1.40 | 0.15 | 1.72 | 3.67 | 0.00 | 1.36 | 1.30 | 5.06 | 12.90 |
| 11 | Piukenai | 2014/06/05 | 12:30 | 4.79 | 4.366 | 0.067 | -73.06 | 1.11 | -11.66 | 0.40 | 13.59 | 0.00 | 13.43 | 8.36 | 0.00 | 2.27 | 6.22 | 13.07 | 12.10 |
| 12 | Tougeshita | 2014/06/05 | 9:30 | 0.31 | 4.065 | 0.066 | -68.01 | 1.55 | -11.37 | 0.44 | 15.12 | 0.00 | 2.90 | 13.12 | 0.00 | 1.23 | 3.91 | 8.00 | 5.46 |
| 13 | Kogen* | 2014/07/26 | 14:00 | N/A | 9.159 | 0.140 | -76.99 | 1.76 | -10.28 | 0.42 | 0.13 | 0.18 | 0.22 | 0.02 | 0.11 | 0.00 | 0.03 | 0.23 | 0.09 |

*indicates rain water sample; N/A – not applicable

**Table 3. Mean transit times (MTTs) estimated using exponential(70%)-piston flow(30%) model (E70%PM) described in the text. One, two or three possible MTTs are obtained using relative error (RE) between analysed and simulated tritium.**

| ID | Location | Date | $^3$H, TU | ±σ, TU | Scaling factor | MTT, yrs | RE, - | MTT, yrs | RE, - | MTT, yrs | RE, - |
|----|----------|------|-----------|--------|----------------|----------|-------|----------|-------|----------|-------|
| | Sample information | | Analyzed | | | 1st solution | | 2nd solution | | 3rd solution | |
| 1a | Otarunai | 2014/06/04 | 4.257 | 0.070 | 2.10 | 14 | 0.003 | - | - | - | - |
| 1b | Otarunai | 2014/10/24 | 4.184 | 0.063 | 2.10 | 14 | 0.003 | - | - | - | - |
| 2 | Takinosawa | 2014/10/24 | 4.114 | 0.062 | 2.10 | 13 | 0.003 | - | - | - | - |
| 3 | Izariirisawa | 2014/10/23 | 3.825 | 0.070 | 2.05 | 13 | 0.048 | - | - | - | - |
| 4 | Honryujyuryu | 2014/10/23 | 3.926 | 0.061 | 2.10 | 13 | 0.046 | - | - | - | - |
| 5 | Kouryu | 2014/10/22 | 3.748 | 0.065 | 2.05 | 13 | 0.043 | - | - | - | - |
| 6 | Hakusen | 2014/10/22 | 4.101 | 0.064 | 2.05 | 14 | 0.000 | - | - | - | - |
| 7 | Okukatsura | 2014/06/04 | 5.290 | 0.086 | 2.15 | 1 | 0.008 | 4 | 0.041 | 23 | 0.002 |
| 8a | Ikutora | 2014/06/06 | 4.659 | 0.077 | 2.10 | 7 | 0.013 | 19 | 0.010 | - | - |
| 8b | Ikutora | 2014/10/03 | 4.449 | 0.065 | 2.10 | 8 | 0.002 | 17 | 0.009 | - | - |
| 9a | Rubeshinai | 2014/06/05 | 4.911 | 0.072 | 2.15 | 7 | 0.016 | 20 | 0.004 | - | - |
| 9b | Rubeshinai | 2014/10/02 | 4.816 | 0.071 | 2.15 | 7 | 0.001 | 20 | 0.003 | - | - |
| 10a | Ishikaridaira | 2014/07/26 | 5.059 | 0.090 | 2.15 | 6 | 0.019 | 22 | 0.007 | - | - |
| 10b | Ishikaridaira | 2014/10/02 | 4.849 | 0.068 | 2.15 | 5 | 0.019 | 22 | 0.004 | - | - |
| 11 | Piukenai | 2014/06/05 | 4.366 | 0.067 | 2.10 | 10 | 0.009 | 16 | 0.009 | - | - |
| 12 | Tougeshita | 2014/06/05 | 4.065 | 0.066 | 2.05 | 11 | 0.003 | 13 | 0.009 | - | - |