# Peer review of "Application of tritium in precipitation and baseflow in Japan: A case study of groundwater transit times and storage in Hokkaido watersheds"

_Hydrology and Earth System Sciences, 2016_

## Referee Comment (RC1) · Anonymous Referee #1 · 21 May 2016

This study presents a new dataset of river water samples that have been analyzed for their oxygen ($\delta18O$) and hydrogen ($\delta2H$, 3H) isotope compositions and their dissolved major ion and nutrient concentrations. The work builds on a strong background of tritium-based explorations developed by several of the coauthors of the manuscript. It is my opinion that some rewording and additional discussion could help this paper, which is already quite strong, to better relate its findings to other partially-overlapping fields, two of which include groundwater storage-depth characterizations, and stable-isotope-based transit time evaluations.

1. Stable O and H isotope versus tritium based approaches: One key and sometimes overlooked issue with the stream water transit time status quo is the roughly orderof-magnitude difference between stable isotope based transit times (reported transit times of a few months up to about five years) and tritium based transit times (reported transit times generally ranging from years to decades; McGuire and McDonnell, 2006). A helpful review of this inter-tracer difference was written by Stewart et al. (2010). Although a time series of stable isotopes was not developed in this study, I think a short discussion about how the storage volumes calculated here compare to stable isotope based storage volumes (e.g., Leopoldo et al. 1992) could benefit the manuscript. Doing so may help to relate the manuscript findings to work completed by research groups publishing rather different mean transit times based on 18O and 2H, plausibly linked to assumptions about age distributions. At a minimum, I think some discussion about the numerous stable isotope based studies of mean transit time with citations to these works could help to better connect these different takes on stream water age.

2. Ambiguity of tritium ages and importance of time-series sampling: I think some statements about the uniqueness of ages and their determination based on a single sample should be softened. Vulnerabilities of stable isotope based mean transit times to aggregation error has been recently discussed by Kirchner et al. (2016a, b). I think that it remains a possibility that tritium based calculations are also susceptible to aggregation error, yielding calculated mean ages that differ substantially from true mean ages. I agree with the authors' statement that a time series of stream tritium could lead to new insights about mean transit times and flow conditions, as it has in their past works (e.g., Morgenstern et al., 2015). However, I think that without these time series data (and perhaps even with these data) there remains at least some room for ambiguous ages, as one could always postulate different mixtures of waters—however unlikely—that yield near-identical tritium concentrations in the mixed sample, but have different true average ages.

3. Framing findings in terms of baseflow (e.g., article title): The authors may, after possible additions or changes resulting from the following point 5 of this review, consider revising the title wording, replacing "groundwater" with "baseflow."

4. Units normalized to catchment area: The groundwater storage volumes reported in the text and in Table 1 could be more straightforwardly compared among the study catchments and with other studies if normalized by catchment area (e.g., point 5 below).

5. Comparing and connecting the calculated groundwater storage with other works: For catchment 1, the reported volume (82.3 million cubic metres of water) and catchment area (104 square kilometres) point to a groundwater storage volume totaling ∼0.8 m. The reported groundwater storage for catchment 1 (0.8m) is more than 100 times smaller than recent estimates of groundwater storage at a global scale (180m; Gleeson et al., 2016), perhaps due to this manuscript's focus on the groundwater that moves into streams as baseflow as the authors do point out. The calculated storage volume is reported to be large on line 18 (pp. 12), but "large" is relative. Juxtaposed against the estimated 180m of groundwater in the upper 2km of the crust, the calculated storage appears rather small. However, on the other hand, the reported catchment 1 storage is more than 10 times larger than terrestrial waters that are stored for less than a few months before entering streams (∼55mm or less; Jasechko et al., 2016). I think that the manuscript's findings may better connect to a broader audience of water scientists that focus on both groundwater and surface water ages if two elements could be added: 5a) a clear and, if possible, quantitative definition of what the storage calculated in the manuscript refers to; and, 5b) further discussion of groundwater/surface water connectivity, groundwater flow velocity with depth and how the storage volumes calculated in this work relate to other published groundwater age and storage estimates

6. Assumptions and limits of the cited and applied transit time model: The lumped parameter model used in this study (Jurgens et al., 2012) can provide a helpful foundation for interpreting tracer measurements. I do suspect that the researchers that developed this model would agree that using the ratio of 70% exponential and 30% piston flow includes assumptions that remain to be validated, and that the model will not characterize flow in all hydrologic settings. For example, other works using different assumptions

about flow (50% exponential flow) have yielded rather impressive matches between modelled and measured spring water tritium (Morgenstern et al., 2015). I recommend a few changes that may help to convey throughout the manuscript that the mean transit times calculated here require assumptions that have not been fully validated. Some spots in the current text where a reminder to readers that the results should be interpreted as estimations include: 6a) pp 1 Line 18 – replace word "determine" with "estimate"; 6b) pp 1 Line 31 – replace words "determine the correct" with "estimate"; 6c) pp 11 Line 18 – add text similar to "if assumptions about age distributions are made" following "Japanese catchments"; 6d) pp 13 Line 24 – add text similar to ", assuming that the 70% exponential and 30% piston flow model applied here describes catchment flow conditions" following "Japanese catchments"; 6e) replace "found unique MTT" with "model a unique MTT"

7. Recent rains and snow: More than one sample was collected from a single river for several study watersheds (1, 8, 9, 10, 11; Table 2 in the manuscript). It appears that most of the paired samples have similar $\delta$18O values (within 0.3 per mille) and similar tritium concentrations (within 0.5 TU) when the two samples collected from one river are compared (sites 1, 8, 9, 10). At site 11 both the measured $\delta$18O value (difference of $\sim$1.5 per mille) and the measured tritium activity (0.7 TU) differ between the June and October samples. It is possible that the observed seasonal difference in $\delta$18O and 3H at this site is related recent precipitation influxes to the river, since precipitation $\delta$18O and 3H vary intra-annually as the manuscript highlights in Figure 5. That the seasonality of river chemistry reflects a damped and phase-shifted precipitation stable isotope cycle has been highlighted by other works (e.g., McGuire and McDonnell, 2006), and a plausible explanation for the published data is that a fraction of the water in the river derives from precipitation that enters the stream quite quickly. Based on the flow model applied to this study, is it possible to include an estimate or to perhaps discuss the possible presence of water in the stream that is much younger than the reported mean transit times? Otherwise, perhaps the addition of a short discussion about intra-annual variability in river isotope compositions that points to the data for stream #11 could be

a useful addition.

8. Calculations of the "average water depth" (e.g., pp. 13 on Line 24) might be better reported as a saturated thickness of water, rather than making an assumption about the porosity of the subsurface. Otherwise, the assumed porosity of 0.1 should be further discussed.

9. Hydrologic model: It is my opinion that this paper could be more cohesive and perceived stronger without the text describing a hydrologic model on page 12 starting at Line 20. If the authors choose to retain this subsection and results, some further description of the model may be useful. For example: why does exactly 20% of precipitation (rain?) and 80% of snow recharge the aquifer? Does the snow recharge the aquifer immediately, or is an energy balance used to model the timing of melt? Does all rain and snow recharge the aquifer or does some runoff? I do appreciate the use of a hydrologic model and its coupling to the analysis of tracer data, but feel that the strongest components of the current manuscript are found in other sections.

10. Minor suggestions:

i) Some of the acronyms used in the study could be somewhat distracting. The authors can consider removing the following acronyms, but this suggestion is, indeed, one of a personal preference for few acronyms: MCM, GNIP (at minimum the "GNIP" in parentheses can removed from the abstract), MAFs, EMM, CDF, EPM, E70%PM.;

ii) Add a citation to earlier works that have used stream water tracers to calculate groundwater storage volume using a similar equation (e.g., Leopoldo et al., 1992);

iii) Superscript Line 3 on pp. 6;

iv) change "amsl" to the more common form "masl," or add units of metres following numeric values in the text;

v) Line 29, pp. 8 "as" to "at";

vi) pp. 10 consider rewording "groundwater watershed";

vii) Line 27, pp. 8 possible rewording from "groundwater transit times" to "baseflow transit times";

viii) Line 17, pp. 11 remove "a";

ix) Line 23 pp. 11 add "or differences in dissolution rates" following "younger MTTs.";

x) Line 28 pp. 11 "volume" to "volumes";

xi) Line 6, pp. 12 " and" after "(#4),";

References:

Gleeson, T., Befus, K., Jasechko, S., Luijendijk, E. and Cardenas, M. B.: The global volume and distribution of modern groundwater. Nature Geoscience, 9, 161–168, 2016.

Jasechko, S., Kirchner, J. W., Welker, J. M. and McDonnell, J. J.: Substantial proportion of global streamflow less than three months old. Nature Geoscience, 9, 126–129, 2016.

Jurgens, B.C., Böhlke, J.K., and Eberts, S.M.: TracerLPM (Version 1): An Excel$^{®}$ workbook for interpreting groundwater age distributions from environmental tracer data. U.S. Geological Survey Techniques and Methods Report 4-F3, 60 p., 2012.

Kirchner, J.W.: Aggregation in environmental systems–Part 1: Seasonal tracer cycles quantify young water fractions, but not mean transit times, in spatially heterogeneous catchments. Hydrol. Earth Syst. Sci, 20, 279–297, 2016a.

Kirchner, J.W.: Aggregation in environmental systems–Part 2: Catchment mean transit times and young water fractions under hydrologic nonstationarity. Hydrol. Earth Syst. Sci, 20, 299–328, 2016b.

Leopoldo, P.R., Martinez, J.C. and Mortatti, J.: Estimation using 18O of the water residence time of small watersheds. In: Proceedings of an International Symposium

on Isotope Techniques in Water Resources Development. International Atomic Energy Agency, Vienna, IAEA-SM-319/4, 75-84, 1992.

McGuire, K.J. and McDonnell, J.J.: A review and evaluation of catchment transit time modeling. J. Hydrol., 330, 543–563, 1992.

Morgenstern, U., Daughney, C.J., Leonard, G., Gordon, D., Donath, F.M. and Reeves, R.: Using groundwater age and hydrochemistry to understand sources and dynamics of nutrient contamination through the catchment into Lake Rotorua, New Zealand. Hydrol. Earth Syst. Sci., 19, 803–822, 2015.

Stewart, M. K., Morgenstern, U. and McDonnell, J. J.: Truncation of stream residence time: how the use of stable isotopes has skewed our concept of streamwater age and origin. Hydrol. Process., 24, 1646–1659, 2010.

---

## Referee Comment (RC2) · Anonymous Referee #2 · 24 May 2016

This work is important to the scientific community as it addresses issues that will be of great importance in the future. They are trying to estimate timescales for residence times of groundwaters in river basins. These groundwaters are important in the maintenance of base flow in the river during all seasons. It is important to have such knowledge of such parameters for proper management of water sheds both in terms of water quantity and quality. They suggest that a small number of tritium measurements can be used to obtain such information and give an example in a study in a Japanese watershed. In general the paper is clear and uses a well-known approach to analyzing tritium data in rivers, i.e. the EPM. Their analysis of the tritium results to determine timescales for the rivers seems to be correct and they do furnish a clear rationale for the conclusions they reach. As far as the quality of the data, the New Zealand laboratory is known for the high quality of their isotopic measurements. One technical point I would make is that they report too many significant figures at times, i.e. the results should be 4.66 +/- 0.07, not 4.659 +/- 0.067. The references list the papers necessary to understand where the model comes from and what they are trying to do with the data. The biggest problem I have with the paper is the input source function, i.e. tritium concentrations in precipitation that are used in the model. The long-term source function is very well constructed with the use of measured data, correlations and concentrations estimates derived from wines. However, with the short timescales for some of the groundwaters in the river basins, results are extremely sensitive to the concentrations in incoming precipitation in the few years just before the stream measurements were made. It is very hard for a reader to know what recent input concentrations are as they use a log scale to address tritium concentrations over the bomb peak period. I think an inset of a secondary graph for the last few years would improve this presentation. It would let the reader know what concentrations they are using in the model for the last few years which is extremely important at the timescales they find in the paper. They clearly understand the importance of the input function by the way they use stable isotopes and other methods to slightly adjust the input function. Two issues are of concern to me. First they only have one precipitation measurement from the time of the study which seems to be higher than what would be expected. Secondly they suggest that some of the issues they have in estimating timescales are caused by snowmelt which is higher in tritium than expected. However, typically in the Northern Hemisphere, tritium concentrations in precipitation are lowest in the winter and during snow accumulation and higher in the spring and summer which appears to be the opposite of what they are suggesting. Unfortunately no snow measurements seem to have been made. They suggest that only one measurement of tritium in stream water is necessary for understanding timescales in watersheds. While this may be true for the stream, it is evident that they should also suggest that measurements of precipitation are important for a correct analysis of the watershed. Overall I would give the paper an excellent for scientific significance, and good for both scientific quality and presentation quality. It gives a good rational for the use of tritium to study the timescales of water within river basins. It also shows that at this stage of the bomb transient, a small number of measurements could yield valuable information for water managers. The relative simplicity and low cost of this approach makes it very desirable.

---

## Author Comment (AC1) · 2 Jun 2016

Manuscript in Review, doi:10.5194/hess-2016-164, 2016

Reply to the Anonymous Referee #2

Comment: This work is important to the scientific community as it addresses issues that will be of great importance in the future. They are trying to estimate timescales for residence times of groundwaters in river basins. These groundwaters are important in the maintenance of base flow in the river during all seasons. It is important to have such knowledge of such parameters for proper management of water sheds both in terms of water quantity and quality. They suggest that a small number of tritium measurements

can be used to obtain such information and give an example in a study in a Japanese watershed. In general the paper is clear and uses a well-known approach to analyzing tritium data in rivers, i.e. the EPM. Their analysis of the tritium results to determine timescales for the rivers seems to be correct and they do furnish a clear rationale for the conclusions they reach. As far as the quality of the data, the New Zealand laboratory is known for the high quality of their isotopic measurements. One technical point I would make is that they report too many significant figures at times, i.e. the results should be 4.66 +/- 0.07, not 4.659 +/- 0.067. The references list the papers necessary to understand where the model comes from and what they are trying to do with the data.

Reply: We thank the Anonymous Referee for providing this positive comment and summarizing the important points of our work. We will reduce the number of significant digits in the text of the manuscript as suggested in the technical point.

Comment: The biggest problem I have with the paper is the input source function, i.e. tritium concentrations in precipitation that are used in the model. The long-term source function is very well constructed with the use of measured data, correlations and concentrations estimates derived from wines. However, with the short timescales for some of the groundwaters in the river basins, results are extremely sensitive to the concentrations in incoming precipitation in the few years just before the stream measurements were made. It is very hard for a reader to know what recent input concentrations are as they use a log scale to address tritium concentrations over the bomb peak period. I think an inset of a secondary graph for the last few years would improve this presentation. It would let the reader know what concentrations they are using in the model for the last few years which is extremely important at the timescales they find in the paper. They clearly understand the importance of the input function by the way they use stable isotopes and other methods to slightly adjust the input function. Two issues are of concern to me. First they only have one precipitation measurement from the time of the study which seems to be higher than what would be expected.

Secondly they suggest that some of the issues they have in estimating timescales are caused by snowmelt which is higher in tritium than expected. However, typically in the Northern Hemisphere, tritium concentrations in precipitation are lowest in the winter and during snow accumulation and higher in the spring and summer which appears to be the opposite of what they are suggesting. Unfortunately no snow measurements seem to have been made. They suggest that only one measurement of tritium in stream water is necessary for understanding timescales in watersheds. While this may be true for the stream, it is evident that they should also suggest that measurements of precipitation are important for a correct analysis of the watershed.

Reply: We thank the Referee for highlighting the importance of the tritium input function. We completely agree that tritium measurements in precipitations are essential for the local tritium studies in Japan and other countries. Having these tritium precipitation measurements provides the site-specific information for scaling of the established input function in many areas. We are now preparing a separate manuscript that discusses construction and scaling of the long-term time-series tritium input function using local data in Japan. For the Hokkaido area, we have collected precipitation and snow core samples for tritium analysis during January-April 2016 at several sites of the Ishikari River basin. These results will be included in a separate publication on the Japanese tritium input in precipitation (Gusyev et al., 2016). From these results we see that the tritium concentrations of snow measurements are higher than the tritium measured at baseflows in Hokkaido. It seems that this statement was not clear in the manuscript and the Referee misunderstood our statement. We will adjust the text in the current manuscript and provide a reference for the manuscript in preparation, see below. To clarify another issue, we collected only one rain water sample during a major rain event in July to compare the tritium concentration in that event with the tritium in the river water. We will adjust the scale of the Figure to include the full range of tritium during the bomb-peak and attempt to include an inset with recent tritium concentrations as commented by the Referee.
References: Gusyev, M.A., Morgenstern, U., Stewart, MK., et al. (2016) Establishing long-term tritium in precipitation input for Japan. Journal of Hydrology, in preparation.

Comment: Overall I would give the paper an excellent for scientific significance, and good for both scientific quality and presentation quality. It gives a good rational for the use of tritium to study the timescales of water within river basins. It also shows that at this stage of the bomb transient, a small number of measurements could yield valuable information for water managers. The relative simplicity and low cost of this approach makes it very desirable.

Reply: We thank the Referee for recognizing the scientific significance of the proposed tritium approach and its practical applications for water resources management in the near future.

---

## Author Comment (AC2) · 14 Jun 2016

Manuscript in Review, doi:10.5194/hess-2016-164, 2016

Reply to the Anonymous Referee #1

Comment: This study presents a new dataset of river water samples that have been analyzed for their oxygen (18O) and hydrogen (2H, 3H) isotope compositions and their dissolved major ion and nutrient concentrations. The work builds on a strong background of tritium-based explorations developed by several of the coauthors of the manuscript. It is my opinion that some rewording and additional discussion could help this paper, which is already quite strong, to better relate its findings to other partiallyoverlapping fields, two of which include groundwater storage-depth characterizations, and stable-isotope-based transit time evaluations.

Reply: We thank the Anonymous Referee for this positive and constructive comment that allows us to highlight the importance of this tritium approach for subsurface characterization and water resources management. We think the implemented changes will have significantly improved our manuscript.

Comment: 1. Stable O and H isotope versus tritium based approaches: One key and sometimes overlooked issue with the stream water transit time status quo is the roughly order-of-magnitude difference between stable isotope based transit times (reported transit times of a few months up to about five years) and tritium based transit times (reported transit times generally ranging from years to decades; McGuire and McDonnell, 2006). A helpful review of this inter-tracer difference was written by Stewart et al. (2010). Although a time series of stable isotopes was not developed in this study, I think a short discussion about how the storage volumes calculated here compare to stable isotope based storage volumes (e.g., Leopoldo et al. 1992) could benefit the manuscript. Doing so may help to relate the manuscript findings to work completed by research groups publishing rather different mean transit times based on 18O and 2H, plausibly linked to assumptions about age distributions. At a minimum, I think some discussion about the numerous stable isotope based studies of mean transit time with citations to these works could help to better connect these different takes on stream water age.

Reply: We thank the Referee for this comment and will include a short discussion about previous studies of mean transit times (MTTs) obtained with tritium and stable isotopes in the "Simulated transit times" section: "We indicate the importance of groundwater storage characterization with tritium river water samples at baseflow by a comparison of stable isotopes and tritium simulated MTTs. Out of seventeen tritium samples, only three samples have MTTs below 5 years at baseflow while modelled MTTs of 12 samples range between 6 and 23 years (Table 1). For these 12 samples, only tritium analysis allows us to characterize groundwater storage with long transit times from years to decades due to the limitation of 18O and 2H stable isotopes for identifying MTTs older than 5 year (McGuire and McDonnell, 2006). This order-of-magnitude difference in sensitivity between the stable isotope and the tritium method will naturally result in that the stable isotope method is preferably applied to short transit time and low volume systems, and the tritium method - to long transit time and large volume systems. Therefore the difference in stable isotope and tritium derived water storages is driven by the difference in MTTs. In addition, the aggregation error proposed by Kirchner (2016a, b) may cause stable isotope derived MTTs to underestimate storage. It has been demonstrated that the use of stable isotopes enables MTT simulation in the range of a few months up to about five years (McGuire and McDonnell, 2006) for groundwater storage volume estimates (Małoszewski et al., 1992; Leopoldo et al., 1998; McGuire et al., 2002; Jasechko et al., 2016). Leopoldo et al. (1998) simulated MTTs of about 0.4 years with 18O values in two Brazilian agricultural watersheds of 1.6 km2 and 3.3 km2 and obtained groundwater volume of 0.1 x 106 m3 with 0.06 m saturated thickness of water for Bufalos watershed and 0.37 x 106 m3 with 0.11 m saturated thickness of water for Paraiso watershed. In cases when simulated MTTs from stable isotopes and tritium have similar values, the groundwater storage volumes do not differ much. For example, Małoszewski et al. (1992) reported similar estimated MTTs of about 4.1 years with 18O and tritium in the Wimbachtal valley watershed of 33 km2 and computed subsurface water volume of 220 x 106 m3 with 6.6 m of saturated thickness of water. MTTs obtained with stable isotope and tritium tracers in many catchments have been summarized by Stewart et al. (2010). Following Kirchner (2016a, b) the vulnerabilities of tritium based MTTs to aggregation error needs to be investigated further."

Comment: 2. Ambiguity of tritium ages and importance of time-series sampling: I think some statements about the uniqueness of ages and their determination based on a single sample should be softened. Vulnerabilities of stable isotope based mean transit times to aggregation error has been recently discussed by Kirchner (2016a, b). I think that it remains a possibility that tritium based calculations are also susceptible

to aggregation error, yielding calculated mean ages that differ substantially from true mean ages. I agree with the authors' statement that a time series of stream tritium could lead to new insights about mean transit times and flow conditions, as it has in their past works (e.g., Morgenstern et al., 2015). However, I think that without these time series data (and perhaps even with these data) there remains at least some room for ambiguous ages, as one could always postulate different mixtures of waters (however unlikely) that yield near-identical tritium concentrations in the mixed sample, but have different true average ages.

Reply: The Referee's comment raises the very important issue of the MTT aggregation error. Kirchner (2016a) discussed the MTT aggregation error of 18O using two neighboring headwater catchments with hypothetical transit times and indicated that tritium-inferred ages should be tested in the same way. The Referee also suggests that interpretation of the tritium data may lead to the same pattern of age aggregation error as shown by Kirchner (2016a). It seems that our Hokkaido results can be used to provide a field example of tritium MTT aggregation. For this comparison we use neighbouring locations in similar hydrogeological settings: Otarunai location (#1) with an area of 68 km2 and Takinosawa (#2) with an area of 44 km2. On October 24th, Otarunai (#1b) had tritium of 4.18 TU at baseflow of 3.66 m3 s-1 and Takinosawa (#2) - 4.11 TU at 0.53 m3 s-1. The simulated MTTs with E70%PM are 14 and 13 years for Otarunai (#1b) and Takinosawa (#2), respectively (Table 1). The combined discharge for these two locations is 4.19 m3 s-1 leading to a tritium concentration of 4.12 TU and aggregated MTT of 13.9 years. The tritium concentration of the aggregated catchments is 4.12 TU giving MTT of 13.6 years using E70%PM. This good agreement of MTTs shows that the MTT aggregation error is very low (about 2%) when combining these waters of these two catchments. The aggregated MTT of 13.6 years is still the only unique best-fit solution in the range of MTTs between 1 and 100 years. This point was illustrated in Figure 8 inset with the one best-fit MTT that can be selected when interpreting tritium values after the disappearance of the Northern Hemisphere bomb-peak tritium (the detailed discussion is provided by Stewart and Morgenstern (2016)). From

3 pairs of catchments in Table 1, we find that neighboring catchments with topographic heterogeneity have low MTT aggregation error when 1) similar tritium concentrations are analyzed at baseflow; 2) one best-fit MTT solution is simulated due to the absence of bomb-peak tritium, and 3) similar transit time distributions of groundwater flow are selected due to hydrogeologic similarity. Once these criteria are violated, the MTT aggregation error of neighboring catchments may be significant. This preliminary finding should be further investigated for other tritium cases in light of the discussion by Kirchner (2016a). We thank the Referee for raising this interesting point and will include a short discussion in the revised manuscript. Clarifying this in detail warrants a separate paper as this is an important issue for groundwater dating.

Comment: 3. Framing findings in terms of baseflow (e.g., article title): The authors may, after possible additions or changes resulting from the following point 5 of this review, consider revising the title wording, replacing "groundwater" with "baseflow."

Reply: We follow the Referee's suggestion and will change "river water" to "baseflow" as it better represents the application of tritium sampling in this study. However, we will keep "groundwater" unchanged to indicate the sources of baseflow and results of transit times estimations and storage characterizations in the subsurface. Replacing "groundwater" by "baseflow" could also mislead the audience by implying that we are using tritium to estimate transit times of river water flows. The new title is as follows: "Application of tritium in precipitation and baseflow in Japan: A case study of groundwater transit times and storage in Hokkaido watersheds"

Comment: 4. Units normalized to catchment area: The groundwater storage volumes reported in the text and in Table 1 could be more straightforwardly compared among the study catchments and with other studies if normalized by catchment area (e.g., point 5 below).

Reply: We follow the Referee's suggestion. The water storage in the five dams is equivalent to average saturated thicknesses of water over each catchment of 0.1 m for Izarigawa Dam, 0.2 for Chubetsu Dam, 0.3 m for Hoheikyo, Katsurazawa and Kanayama Dams, 0.4 m for Taisetsu Dam, 0.6 m for Rumoi Dam and 0.8 m for Jyozankei Dam, see our reply #5. We describe this as follows. "The importance of the subsurface groundwater storages for the management of water resources can also be emphasized by comparing them with the normalized storages of the five dams (i.e. water storage in the reservoir divided by the corresponding catchment area) (Table 1). For these five dams, this average saturated thickness of water ranges between 0.1 and 0.8 m and is much smaller than storage in the study headwater catchments, which have the saturated thicknesses of water between 0.19 and 24 m."

Comment: 5. Comparing and connecting the calculated groundwater storage with other works: For catchment 1, the reported volume (82.3 million cubic metres of water) and catchment area (104 square kilometres) point to a groundwater storage volume totaling 0.8 m. The reported groundwater storage for catchment 1 (0.8m) is more than 100 times smaller than recent estimates of groundwater storage at a global scale (180m; Gleeson et al., 2016), perhaps due to this manuscript's focus on the groundwater that moves into streams as baseflow as the authors do point out. The calculated storage volume is reported to be large on line 18 (pp. 12), but "large" is relative. Juxtaposed against the estimated 180m of groundwater in the upper 2km of the crust, the calculated storage appears rather small. However, on the other hand, the reported catchment 1 storage is more than 10 times larger than terrestrial waters that are stored for less than a few months before entering streams (55mm or less; Jasechko et al., 2016). I think that the manuscript's findings may better connect to a broader audience of water scientists that focus on both groundwater and surface water ages if two elements could be added: 5a) a clear and, if possible, quantitative definition of what the storage calculated in the manuscript refers to; and, 5b) further discussion of groundwater/surface water connectivity, groundwater flow velocity with depth and how the storage volumes calculated in this work relate to other published groundwater age and storage estimates.

Reply: We will follow the Referee's suggestion and adopt the suggested changes. To address the Referee's comment 5a), we add that we computed the subsurface volume of the groundwater system contributing to the baseflow. This subsurface volume provides the majority of baseflow especially during winter conditions in Hokkaido. It is possible that this groundwater volume could be further divided into shallow and deeper components of groundwater storage. However, this task is beyond the scope of our study. For Referee's comment 5b), we will enhance the discussion of Hokkaido groundwater storage as well as the average saturated thickness of water obtained at baseflow, while limiting discussion of groundwater/surface water connectivity and groundwater velocities to a short statement because there was only limited field data. The field data had been obtained from hydrogeological studies at several dam construction sites. It seems that the dam storage values reported in Table 1 were misinterpreted by the Referee. In Table 1, we provide the drainage area and capacity of dams that are located downstream of our study sites to indicate the importance of subsurface storage. Therefore, catchment #1 in the Referee's comment refers to the storage and drainage area of Jyozankei Dam that is located downstream of Otarunai and Takinosawa locations. We clarified this point and introduced more information about estimated groundwater storage, see below: "For the Otarunai and Takinosawa locations, we used MTTs of 13 and 14 years with baseflow values of 3.66 and 0.53 $m^3$ $s^{-1}$ to find groundwater storage of 1616 and 217 x 106 $m^3$, respectively. Dividing these two volumes by the respective drainage areas of 64 and 14 $km^2$ in Table 1 we find the saturated thickness of water of 24 m for Otarunai and 16 m for Takinosawa. These values of saturated water thickness are about 10 times smaller than the recent estimates of groundwater storage thickness of 180 m by Gleeson et al. (2016). For nearby catchments the saturated water thickness of the Izariirisawa location with catchment area of 42 $km^2$ is 6.9 m (estimated from 291 x 106 $m^3$ storage based on MTT of 13 years and 0.71 $m^3$ $s^{-1}$ baseflow). The Honryujyuryu location has 15 m saturated water thickness (estimated from 947 x 106 $m^3$ storage obtained at 2.3 $m^3$ $s^{-1}$ baseflow and catchment area of 65 $km^2$). The saturated water thickness of Ishikaridaira location is about 24 m (estimated from 2720

x 106 m3 storage obtained using MTT of 22 years and catchment area of 113 km2), while the Rubeshinai location has 4.9 m saturated thickness of water (from an area of 45 km2 and 334 x 106 m3 storage). The Ikutora location has the largest drainage area of 377 km2 and saturated water thickness of about 13 m (estimated from 5074 x 106 m3 storage using MTT of 17 years at 9.5 m3 s-1 baseflow). The Tougeshita location has the saturated thickness of water of 1.4 m (from catchment area of 49 km2 and 92 x 106 m3 of storage). For the study site with younger waters, we found the saturated water thickness of 0.19 and 0.76 m for the Okukatsura location with the catchment area of 56 km2 and 13 and 56 x 106 m3 volumes using MTTs of 1 and 4 years. These values of saturated water thickness are only 4 times larger than the saturated water thickness of young (MTT of 0.2 years) terrestrial water identified by Jasechko et al. (2016)."

Comment: 6. Assumptions and limits of the cited and applied transit time model: The lumped parameter model used in this study (Jurgens et al., 2012) can provide a helpful foundation for interpreting tracer measurements. I do suspect that the researchers that developed this model would agree that using the ratio of 70% exponential and 30% piston flow includes assumptions that remain to be validated, and that the model will not characterize flow in all hydrologic settings. For example, other works using different assumptions about flow (50% exponential flow) have yielded rather impressive matches between modelled and measured spring water tritium (Morgenstern et al., 2015). I recommend a few changes that may help to convey throughout the manuscript that the mean transit times calculated here require assumptions that have not been fully validated. Some spots in the current text where a reminder to readers that the results should be interpreted as estimations include: 6a) pp 1 Line 18 – replace word "determine" with "estimate"; 6b) pp 1 Line 31 – replace words "determine the correct" with "estimate"; 6c) pp 11 Line 18 – add text similar to "if assumptions about age distributions are made" following "Japanese catchments"; 6d) pp 13 Line 24 – add text similar to ", assuming that the 70% exponential and 30% piston flow model applied here describes catchment flow conditions" following "Japanese catchments"; 6e) replace "found unique MTT" with "model a unique MTT".

Reply: We agree with the Referee's comment and will implement the changes in 6a-d) as suggested. We will also add a sentence indicating that the ratio of 70% exponential and 30% piston flow was used following Morgenstern et al. (2010) which showed that the piston flow component in this catchment due to flow through the unsaturated zone alone contributes >20% of piston flow already. 30% therefore seems a realistic value.

Comment: 7. Recent rains and snow: More than one sample was collected from a single river for several study watersheds (1, 8, 9, 10, 11; Table 2 in the manuscript). It appears that most of the paired samples have similar 18O values (within 0.3 per mille) and similar tritium concentrations (within 0.5 TU) when the two samples collected from one river are compared (sites 1, 8, 9, 10). At site 11 both the measured 18O value (difference of 1.5 per mille) and the measured tritium activity (0.7 TU) differ between the June and October samples. It is possible that the observed seasonal difference in 18O and 3H at this site is related recent precipitation influxes to the river, since precipitation 18O and 3H vary intra-annually as the manuscript highlights in Figure 5. That the seasonality of river chemistry reflects a damped and phase-shifted precipitation stable isotope cycle has been highlighted by other works (e.g., McGuire and McDonnell, 2006), and a plausible explanation for the published data is that a fraction of the water in the river derives from precipitation that enters the stream quite quickly. Based on the flow model applied to this study, is it possible to include an estimate or to perhaps discuss the possible presence of water in the stream that is much younger than the reported mean transit times? Otherwise, perhaps the addition of a short discussion about intra-annual variability in river isotope compositions that points to the data for stream #11 could be a useful addition.

Reply: We thank the Referee for identifying this point. After the Referee highlighted this issue, we investigated the sample #11b in question and found that not only tritium and the stable isotopes, but also the chemistry of sample #11b is very different to that of #11a in Table 1. Sample #11b was, in contrast to the other samples, not sampled by the authors but by local officers of Chubetsu Dam and we are now almost certain that

a different location has been sampled. Therefore, we excluded results of sample #11b from the manuscript.

Comment: 8. Calculations of the "average water depth" (e.g., pp. 13 on Line 24) might be better reported as a saturated thickness of water, rather than making an assumption about the porosity of the subsurface. Otherwise, the assumed porosity of 0.1 should be further discussed.

Reply: We followed the Referee's suggestion and use "the saturated thickness of water" in the revised text, see our reply to Comment 5. We also add the definition of the saturated water thickness, which is a baseflow times mean transit time of baseflow divided by catchment area.

Comment: 9. Hydrologic model: It is my opinion that this paper could be more cohesive and perceived stronger without the text describing a hydrologic model on page 12 starting at Line 20. If the authors choose to retain this subsection and results, some further description of the model may be useful. For example: why does exactly 20% of precipitation (rain?) and 80% of snow recharge the aquifer? Does the snow recharge the aquifer immediately, or is an energy balance used to model the timing of melt? Does all rain and snow recharge the aquifer or does some runoff? I do appreciate the use of a hydrologic model and its coupling to the analysis of tracer data, but feel that the strongest components of the current manuscript are found in other sections.

Reply: We thank the Referee for this comment. The purpose of our model is a demonstration of a simple calculation approach of groundwater storage change for water resources management upstream of tritium river sampling locations. In this approach, the lumped model does not include any sophisticated calculations such as energy balance, delay in recharge, soil types, etc., and only simulates the changes of saturated groundwater storage that receives recharge from infiltrated soil water and contributes to the baseflow discharge. In our model, we obtained these recharge rates from a range of field values reported by Iwata et al. (2010) for the Tokachi site in Hokkaido.

Iwata et al. (2010) investigated water infiltration rates at 0.2 and 1.05 m soil depth from 2002 to 2006 and reported the largest rates of soil water infiltration during the spring snow melt season between 79% and 85% than the summer-fall water infiltration rates of 20-25% in 2002. Therefore, we included these statements in the manuscript and decided to keep this model discussion. We will add additional information about the utilized model in a separate sub-section "Simulated groundwater storage" and will also include related information provided in our replies #1 and #5. We plan to apply detailed numerical simulations in the next phase of this study.

Comment: 10. Minor suggestions: i) Some of the acronyms used in the study could be somewhat distracting. The authors can consider removing the following acronyms, but this suggestion is, indeed, one of a personal preference for few acronyms: MCM, GNIP (at minimum the "GNIP" in parentheses can removed from the abstract), MAFs, EMM, CDF, EPM, E70%PM.

Reply: We will follow Referee's suggestion to reduce the use of acronyms in the manuscript.

ii) Add a citation to earlier works that have used stream water tracers to calculate groundwater storage volume using a similar equation (e.g., Leopoldo et al., 1992);

Reply: We added this and other references to the manuscript, see our reply to the Referee's comment #1.

iii) Superscript Line 3 on pp. 6;

Reply: We adjusted the text.

iv) change "amsl" to the more common form "masl," or add units of metres following numeric values in the text;

Reply: We replaced "amsl" to "masl" in the text as suggested.

v) Line 29, pp. 8 "as" to "at";

Reply: We replaced "as" by "at" as suggested.

vi) pp. 10 consider rewording "groundwater watershed";

Reply: We replaced "groundwater watershed" by "subsurface groundwater storage".

vii) Line 27, pp. 8 possible rewording from "groundwater transit times" to "baseflow transit times";

Reply: The indicated statement is not available at the Referee's specified location.

viii) Line 17, pp. 11 remove "a";

Reply: We removed "a" as suggested.

ix) Line 23 pp. 11 add "or differences in dissolution rates" following "younger MTTs.";

Reply: We added the text as suggested.

x) Line 28 pp. 11 "volume" to "volumes";

Reply: We changed to "volumes" as suggested.

xi) Line 6, pp. 12 " and" after "(#4),";

Reply: We added "and" as suggested.

References:

Iwata Y., Hirota, T., Hayashi M., Suzuki S., and Hasegawa, S.: Effects of frozen soil and snow cover on cold-season soil water dynamics in Tokachi, Japan. Hydrol. Process. 24, 1755–1765, doi: 10.1002/hyp.7621, 2010.

Małoszewski, P. and Zuber, A.: Determining the turnover time of groundwater systems with the aid of environmental tracers 1. Models and their applicability, J. of Hydrol., 57, 207-231, doi:10.1016/0022-1694(82)90147-0, 1982.

Małoszewski, P, Rauert, W., Trimborn, P., Herrmann, A., and Rau, R.: Isotopoe hydrological study of mean transit times in an alpine basin (Wimbechtal, Germany). Journal of Hydrology, 140, 343-360, 1983.

McGuire, K.J, DeWalle, D.R, and Gburek, W.J.: Evaluation of mean residence time in subsurface waters using oxygen-18 fluctuations during drought conditions in mid-Appalachians. Journal of Hydrology 261, 132-149, 2006.

Morgenstern, U., Stewart, M.K., and Stenger, K.: Dating of streamwater using tritium in a post nuclear bomb pulse world: continuous variation of mean transit time with streamflow. Hydrol. Earth Syst. Sci., 14, 2289-2301, doi:10.5194/hess-14-2289-2010, 2010.

Stewart, M.K. and Morgenstern, U.: Importance of tritium-based transit times in hydrological systems. Wiley Interdisciplinary Reviews: Water, 3(2), 145-154; doi: 10.1002/wat2.1134, 2016.

Gleeson, T., Befus, K., Jasechko, S., Luijendijk, E. and Cardenas, M. B.: The global volume and distribution of modern groundwater. Nature Geoscience, 9, 161–168, 2016.

Jasechko, S., Kirchner, J.W., Welker, J. M. and McDonnell, J. J.: Substantial proportion of global streamflow less than three months old. Nature Geoscience, 9, 126–129, 2016.

Jurgens, B.C., Böhlke, J.K., and Eberts, S.M.: TracerLPM (Version 1): An Excel[®] workbook for interpreting groundwater age distributions from environmental tracer data. U.S. Geological Survey Techniques and Methods Report 4-F3, 60 p., 2012.

Kirchner, J.W.: Aggregation in environmental systems–Part 1: Seasonal tracer cycles quantify young water fractions, but not mean transit times, in spatially heterogeneous catchments. Hydrol. Earth Syst. Sci, 20, 279–297, 2016a.

Kirchner, J.W.: Aggregation in environmental systems–Part 2: Catchment mean transit times and young water fractions under hydrologic nonstationarity. Hydrol. Earth Syst. Sci, 20, 299–328, 2016b.

Leopoldo, P.R., Martinez, J.C. and Mortatti, J.: Estimation using 18O of the water residence time of small watersheds. In: Proceedings of an International Symposium on Isotope Techniques in Water Resources Development. International Atomic Energy Agency, Vienna, IAEA-SM-319/4, 75-84, 1992.

McGuire, K.J. and McDonnell, J.J.: A review and evaluation of catchment transit time modeling. J. Hydrol., 330, 543–563, 2006.

Morgenstern, U., Daughney, C.J., Leonard, G., Gordon, D., Donath, F.M. and Reeves, R.: Using groundwater age and hydrochemistry to understand sources and dynamics of nutrient contamination through the catchment into Lake Rotorua, New Zealand. Hydrol. Earth Syst. Sci., 19, 803–822, 2015.

Stewart, M. K., Morgenstern, U. and McDonnell, J. J.: Truncation of stream residence time: how the use of stable isotopes has skewed our concept of streamwater age and origin. Hydrol. Process., 24, 1646–1659, 2010.